# TELL-TALE: TASK EFFICIENT LLMS WITH TASK AWARE LAYER ELIMINATION

## ABSTRACT

This paper introduces TALE, Task-Aware Layer Elimination, an inference-time algorithm that prunes entire transformer layers in an LLM by directly optimizing task-specific validation performance without retraining. We evaluate TALE on 9 tasks and 5 models, LLaMA 3.1 8B, Qwen 2.5 7B, Qwen 2.5 0.5B, Mistral 7B, and Lucie 7B, under both zero-shot and few-shot settings; and we show that TALE compares favorably to prior approaches, most of which require retraining. Providing user control over trade-offs between accuracy and efficiency, TALE 's selective layer removal consistently improves accuracy while reducing computational cost across all benchmarks. TALE produces additional performance gains when combined with fine-tuning. Analysis shows that certain layers act as bottlenecks, degrading task-relevant representations. TALE remedies this problem, producing smaller, faster, and more accurate models that are also faster to fine-tune while offering new insights into transformer interpretability.

## 1    INTRODUCTION

While Large Language Models (LLMs) have achieved great success, their substantial computational demands prevent resource-constrained organizations and those with high-throughput applications from leveraging more capable models. The use of multi-agent systems, where each agent requires an LLM specialized for a particular role, has intensified the need for methods that simultaneously boost task-specific performance and reduce computation costs. Fine-tuning can increase task performance but does not reduce inference costs and requires significant training overhead and data. General pruning reduces computation costs but typically demands significant retraining and often results in substantial performance degradation on downstream tasks.

We offer TALE , Task Aware Layer Elimination, a method that both **increases task performance and reduces computational overhead**. TALE is a lightweight, greedy, iterative layer pruning algorithm. It operates at inference time, is hardware agnostic, directly optimizes for task-specific accuracy at each pruning step and consistently offers improved results over the original model. This improvement persists in interactions with fine tuning on our tasks. As illustrated in Figure 8 and detailed in Section 3, TALE systematically evaluates all possible single-layer removals at each iteration, selecting the layer whose elimination results in the highest validation accuracy. This process continues iteratively until performance improvements fall below a predefined threshold, ensuring that only layers with minimal or negative impact on task performance are removed.

TALE is based on our observation, illustrated in Figure 1, that not all layers in a transformer contribute to a particular task and indeed sometimes hamper task specific performance. TALE leverages the modular nature of transformer architectures, where each layer performs a complete transformation of the input representation through attention and feedforward mechanisms. This architectural property enables the removal of entire layers without requiring modifications to the remaining network structure. By selectively removing transformer layers, TALE improves task specific accuracy and provides moderate computational reductions with minimal implementation complexity.

We provide experimental evidence that TALE provides consistent improvements in both accuracy and computational efficiency on five LLMs, LLaMA 3.1 8B, Qwen 2.5 7B, Qwen 2.5 0.5B, Mistral 7B and Lucie 7B, on 9 diverse benchmark datasets (Sections 4 and 5) both in zero-shot and few-shot settings. Comparing TALE with previous pruning methods shows that TALE achieves substantially higher accuracy. We also show that pruning with TALE can combine with fine-tuning to provide

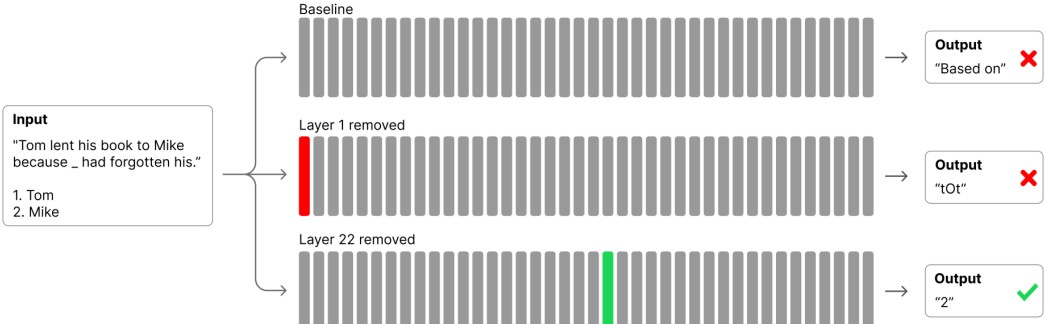

Figure 1: Figure illustrates how TALE improves performance on the Winogrande task in LLaMA 2 13B with 5-shot prompting. The full model hallucinates the answer; random layer deletion leads to nonsensical output; using TALE to remove a layer on the first iteration yields the right prediction.

even greater accuracy gains. We analyze layer flow using the notion of mutual information (MI) to support the hypothesis (Section 5) that not all model layers serve a useful purpose in a given task and may even impede performance, thus challenging the conventional assumption that deeper models necessarily perform better. Additionally, our experiments show TALE's potential as a tool for understanding layer function in and across models (Section 6), thereby aiding model interpretability.

## 2 RELATED WORK

Zhu et al. (2024) distinguishes four primary approaches to reducing model size and computation complexity: model pruning, quantization, low-rank approximation, and knowledge distillation. Our work focuses on pruning, which comprises unstructured, structured, and semi-structured methods. Unstructured pruning removes individual parameters, resulting in irregular, sparse structures Han et al. (2015b); Chen et al. (2015); Srinivas & Babu (2015); structured pruning eliminates entire components such as neurons, attention heads, or layers while maintaining the overall network structure He et al. (2017); Voita et al. (2019); Lagunas et al. (2021); Men et al. (2024). Semi-structured pruning combines fine-grained control with structural regularity, and has been explored in recent work Li et al. (2023); Frantar & Alistarh (2023b); Sun et al. (2024). Early pruning methods leveraged second-order information for structured pruning LeCun et al. (1989); Hassibi et al. (1993), but the field has since shifted toward computationally simpler, magnitude-based approaches that prune parameters by importance scores Han et al. (2015a); See et al. (2016); Narang et al. (2017). Model pruning has also benefited from information-theory (Tishby et al., 2000; Tishby & Zaslavsky, 2015; Ganesh et al., 2020; Westphal et al., 2024). Fan et al. (2021) propose a layer-wise strategy that leverages mutual information estimates to reduce hidden dimensionality in a top-down manner. A central challenge, however, is the difficulty of estimating MI. Despite interesting theoretical work as in Ishmael Belghazi et al. (2018), in practice, probing classifiers Belinkov (2022) remain the dominant tool due for such estimations.

For large transformers, Zhang & Papyan (2025) proposes a pruning strategy using matrix approximations. Similarly, Xia et al. (2023) shows that structured layer and hidden-dimension pruning can create smaller submodels that outperform same-sized models trained from scratch, though they do not match the original model's performance. Kim et al. (2024) explores lock-level pruning based on weight importance. These methods generally require fine-tuning to recover accuracy and are prone to degradation, often needing additional retraining Xia et al. (2024), with improvements typically measured relative to small models rather than the original unpruned baselines.

Closer to TALE are pruning approaches that do not require retraining. Frantar & Alistarh (2023a); Zhang et al. prune contiguous blocks, especially in attention layers, with minimal performance loss. SLEB Song et al. (2024) removes entire layers based on the cosine similarity of their representations, but evaluates perplexity before permanently pruning to avoid degrading linguistic performance. SliceGPT Ashkboos et al. (2024) prunes layer dimensions via Principal Component Analysis, eliminating less informative components in embeddings and hidden states. SparseGPT Frantar & Alistarh (2023c) introduces sparsity by setting individual weights to zero using a reconstruction-

based criterion, while Wanda Sun et al. (2023) removes weights according to the product of their magnitudes and input activation norms.

Although these training-free pruning methods are designed to be general, they often degrade linguistic and reasoning abilities. TALE applies task-specific pruning, optimizing the model for a particular task, which not only improves performance over the original model but also increases inference speed.

## 2.1 BASICS AND INTUITIONS

A transformer maps a sequence of input vectors $(x_1, \cdots, x_n)$ to a corresponding sequence of output vectors through a stack of L layers. Each layer $\ell$ transforms the hidden representations $X^{(\ell)} = (x_1^{(\ell)}, \ldots, x_n^{(\ell)})$ into $X^{(\ell+1)}$ through attention and feedforward blocks, connected by residual pathways. Removing layer $\ell$ from this pipeline simply redirects the flow such that $X^{(\ell-1)} \rightarrow X^{(\ell+1)}$, a property that makes the architecture naturally amenable to layer-wise pruning.

Our initial intuition for TALE came from examining the behavior of partial forward passes. Let $h^{(k)}$ denote the hidden representation after $k$ layers. Instead of always decoding from the final representation $h^{(L)}$, we projected intermediate representations $h^{(k)}$ for $k < L$ directly into the vocabulary space using the output projection $W_{\text{out}}$, i.e.,

$$\hat{y}^{(k)} = \text{softmax}(W_{\text{out}} h^{(k)}).$$

We then compared the performance of $\hat{y}^{(k)}$ across different values of $k$. Surprisingly, we observed that for many tasks, intermediate layers ($k < L$) achieved higher accuracy than the final layer L (Figure 4). This indicated that additional depth does not always translate into better task-specific performance: some layers contribute marginally, while others introduce representational noise.

This experiment led to our central hypothesis: *not all layers in an LLM are equally useful, and selectively removing redundant layers can preserve—or even improve—downstream accuracy*. TALE (Task-Aware Layer Elimination) formalizes this intuition into a principled, iterative pruning strategy.

---

**Algorithm 1** TALE : Greedy Iterative Layer Pruning

**Require:** Pre-trained model $\mathcal{M}$ with $L$ layers; validation set $\mathcal{D}_{val}$; performance threshold $\epsilon$
**Ensure:** Compressed model $\mathcal{M}^*$
1: Initialize $\mathcal{M}^* \leftarrow \mathcal{M}$
2: **repeat**
3:     **for** each layer $\ell \in \{1, \ldots, L\}$ of $\mathcal{M}^*$ **do**
4:         Construct candidate model $\mathcal{M}_{-\ell}$ by removing layer $\ell$
5:         Compute validation accuracy $A_\ell = \text{Acc}(\mathcal{M}_{-\ell}, \mathcal{D}_{val})$
6:     **end for**
7:     Select $\ell^* = \arg\max_\ell A_\ell$
8:     **if** $A_{\ell^*} \geq \text{Acc}(\mathcal{M}^*, \mathcal{D}_{val}) - \epsilon$ **then**
9:         Update $\mathcal{M}^* \leftarrow \mathcal{M}_{-\ell^*}$
10:     **else**
11:         **break**
12:     **end if**
13: **until** All Accuracies below threshold
14: **return** $\mathcal{M}^*$

---

## 2.2 TALE

TALE is a greedy iterative layer pruning algorithm for pre-trained open-weights LLM compression that systematically removes layers while preserving or even improving model performance (Algorithm 6). Starting with a full pre-trained model, TALE evaluates all possible single-layer removals at each iteration, computing the validation accuracy for each candidate pruned architecture. The layer whose removal results in the highest accuracy is permanently eliminated from the model, and this compressed architecture becomes the baseline for the next iteration. This process continues iteratively until the performance improvement falls below a predefined threshold, at which point the

algorithm terminates and returns the most compressed model that maintains performance above the specified threshold. We prune on a subset of a benchmark, while evaluation uses a separate subset within the same distribution. Thus, TALE improves performance on the underlying task itself, rather than merely being specific to the pruning data. Our approach directly optimizes for task-specific accuracy at each pruning step, ensuring that only layers with minimal impact on the target objective are removed. This exhaustive evaluation strategy, while computationally intensive during the pruning phase, provides strong empirical guarantees about the optimality of each pruning decision within the greedy framework.

## 3 BENCHMARKS AND DATASETS

We evaluate TALE across a diverse suite of nine benchmarks spanning reasoning, language understanding, and commonsense knowledge. For mathematical reasoning, we include **GSM8K-Hard**, a curated subset of GSM8K Cobbe et al. (2021) with more than five premises per question to increase difficulty, and **MATH500** Hendrycks et al. (2021b), a benchmark for symbolic and arithmetic reasoning (for evaluation details see Appendix A). For language understanding, we consider **MMLU** Hendrycks et al. (2021a) and **BoolQ** Clark et al. (2019), while **Winogrande** Sakaguchi et al. (2021), **CommonsenseQA** Talmor et al. (2019), and **BIG-Bench** Srivastava et al. (2023) capture commonsense and multi-task generalization. Finally, we include both **ARC-Easy** and **ARC-Challenge** Clark et al. (2018), which evaluate scientific and factual reasoning at varying difficulty levels. Together, these nine datasets cover a broad spectrum of downstream challenges and allow us to assess both the generality and task-specific benefits of our pruning strategy.

## 4 RESULTS

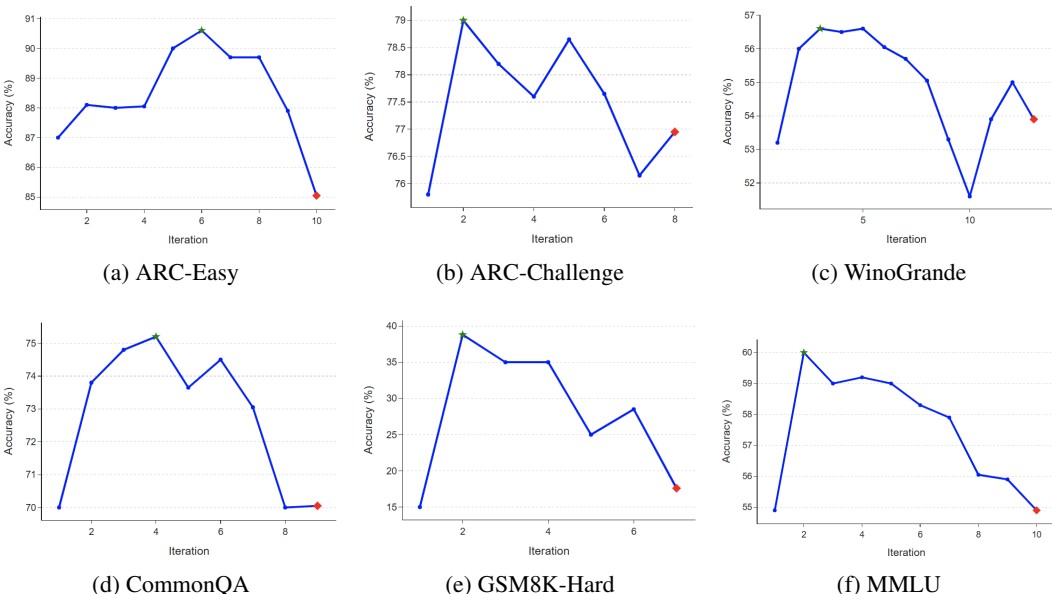

Figure 2: Accuracy progression of TALE across 6 benchmark datasets for LLaMA 3.1 8B. Each curve represents the accuracy at successive iterations. The ⋆ denotes the best-performing layer drop configuration, while the □ highlights the Best Speed up with at least Baseline Accuracy (BSBA) configuration. Plots for all tasks are in Appendix E.

We evaluate TALE across five medium-scale models (LLaMA 3.1 8B, Mistral 7B, Lucie 7B, Qwen 2.5 7B) and one smaller model (Qwen 2.5 0.5B), spanning nine benchmarks that cover commonsense reasoning, reading comprehension, and mathematical problem solving. All experiments are conducted in the zero-shot setting unless otherwise noted.[1]

---

[1]Code available at https://anonymous.4open.science/r/tale/

We employed two evaluation strategies, the standard one from the LM-Eval library (Table 2) and an automatic evaluation (**Our Eval**) that we developed for the test portions of our datasets (see Figure 5, Tables 1, 3, and additional tables in the appendix). The LM-Eval method selects the answer with the highest probability from options provided. This has drawbacks which we discuss in Appendix A and doesn't really measure actual, generated output, whereas Our Eval does. We force the model to predict its answer after reasoning steps in a particular format and then calculate the accuracy. This can lower accuracy from what is expected; but since we used the same evaluation criteria for all the techniques and models and are interested in relative changes in performance under pruning, these unexpected increases/decreases are moot. Table 5 summarizes model configurations.

TALE requires only modestly-sized validation sets for task-specific optimization, ranging from 500 to 1500 examples. As seen in Table 13 (Appendix H), once the validation set size exceeds 500 examples, the set of layers dropped stabilizes across all tasks.

| Dataset | LLaMA 3.1 8B (zero-shot) | | | | | | | Qwen 2.5 7B (zero-shot) | | | | | | |
| | Baseline | Best Model | | | BSBA | | | Baseline | Best Model | | | BSBA | | |
| | Perf. | Perf. | #D | Sp. | Perf. | #D | Sp. | Perf. | Perf. | #D | Sp. saved | Perf. | #D | Sp. |
|---|---|---|---|---|---|---|---|---|---|---|---|---|---|---|
| ARC-Easy | 87.00 | **90.55**(+4.08% ↑) | 5 | -14.6% | 87.82 | 8 | -23.5% | 91.01 | **91.82**(+0.89% ↑) | 2 | -10.0% | 90.91 | 5 | -30.3% |
| ARC-Challenge | 75.86 | **78.62**(+3.63% ↑) | 4 | -11.7% | 76.90 | 7 | -20.5% | 86.55 | **92.00**(+6.45% ↑) | 2 | -6.7% | 86.55 | 6 | -19.9% |
| BoolQ | 85.00 | **86.20**(+1.40% ↑) | 3 | -8.8% | 85.70 | 7 | -17.6% | 84.10 | **86.90**(+3.22% ↑) | 4 | -13.3% | 82.70 | 5 | -23.2% |
| MMLU | 54.87 | **59.90**(+9.17% ↑) | 1 | -2.9% | 54.87 | 9 | -26.4% | 68.10 | **71.00**(+4.26% ↑) | 5 | -16.6% | 68.13 | 6 | -19.9% |
| CommonQA | 72.20 | **75.30**(+4.29% ↑) | 3 | -8.8% | 73.10 | 6 | -17.6% | 80.30 | **84.40**(+5.11% ↑) | 2 | -6.6% | 80.50 | 6 | -19.9% |
| Winogrande | 53.83 | **56.67**(+5.28% ↑) | 4 | -11.7% | 53.83 | 12 | -32.2% | 62.04 | **67.25**(+8.40% ↑) | 3 | -10.0% | 62.19 | 6 | -19.9% |
| BIG-Bench | 75.20 | **83.60**(+11.17% ↑) | 5 | -14.4% | 75.20 | 11 | -32.2% | 79.20 | **81.60**(+3.03% ↑) | 6 | -19.9% | 81.60 | 6 | -19.9% |
| GSM8K-HARD | 15.07 | **37.08**(+146.05% ↑) | 1 | -2.9% | 35.0 | 4 | -11.7% | 7.9 | **27.0**(+243.58% ↑) | 2 | -6.6% | 19.1 | 4 | -13.3% |
| Math500 | 20.50 | **26.00**(+26.83% ↑) | 1 | -2.9% | 26.00 | 3 | -8.8% | 18.00 | **27.00**(+50.0% ↑) | 2 | -6.6% | 21.00 | 4 | -13.3% |

| Dataset | Lucie 7B (zero-shot) | | | | | | | Mistral 7B (zero-shot) | | | | | | |
| | Baseline | Best Model | | | BSBA | | | Baseline | Best Model | | | BSBA | | |
| | Perf. | Perf. | #D | Sp. | Perf. | #D | Sp. | Perf. | Perf. | #D | Sp. | Perf. | #D | Sp. |
|---|---|---|---|---|---|---|---|---|---|---|---|---|---|---|
| ARC-Easy | 72.45 | **76.55**(+5.66% ↑) | 6 | -18.1% | 72.55 | 13 | -39.2% | 81.02 | **83.45**(+4.23% ↑) | 5 | -15.4% | 81.09 | 9 | -27.7% |
| ARC-Challenge | 48.00 | **53.79**(+12.06% ↑) | 7 | -21.1% | 51.38 | 11 | -33.1% | 72.20 | **74.83**(+3.64% ↑) | 6 | -18.5% | 72.41 | 8 | -24.6% |
| BoolQ | 53.70 | **77.50**(+44.32% ↑) | 5 | -17.2% | 60.60 | 19 | -54.2% | 80.36 | **83.20**(+3.53% ↑) | 6 | -18.5% | 80.60 | 10 | -27.7% |
| MMLU | 21.36 | **42.98**(+101.2% ↑) | 8 | -24.1% | 39.39 | 15 | -45.2% | 52.73 | **57.81**(+9.63% ↑) | 2 | -6.2% | 52.91 | 8 | -24.6% |
| CommonQA | 55.50 | **69.70**(+25.59% ↑) | 3 | -9.1% | 57.10 | 17 | -48.2% | 57.32 | **61.40**(+7.12% ↑) | 4 | -12.3% | 57.40 | 7 | -21.5% |
| Winogrande | 54.20 | **57.80**(+6.64% ↑) | 5 | -27.1% | 54.30 | 15 | -45.2% | 52.55 | **58.80**(+11.53% ↑) | 10 | -30.7% | 53.43 | 13 | -40.0% |
| BIG-Bench | 69.60 | **77.20**(+9.84% ↑) | 9 | -27.1% | 72.00 | 15 | -45.1% | 70.00 | **76.40**(+9.14% ↑) | 9 | -28.0% | 72.80 | 11 | -33.8% |
| GSM8K-HARD | 14.20 | **17.80**(+25.35% ↑) | 1 | -3.1% | 17.40 | 3 | -9.1% | 11.24 | **19.10**(+69.92% ↑) | 2 | -6.2% | 15.73 | 4 | -12.3% |
| Math500 | 19.00 | **27.00**(+42.11% ↑) | 2 | -6.0% | 26.00 | 3 | -9.1% | 8.00 | **16.00**(+100% ↑) | 1 | -3.1% | 10.00 | 4 | -12.3% |

Table 1: Performance comparison across language models under 0-shot evaluation. Accuracy (**Perf.**) uses Our Eval We also report number of dropped layers (**#D**), and relative inference speedup (**Sp.**) in terms of percentage of Tflops saved (Percentage saved = $\frac{\text{Tflops}_{\text{Baseline}} - \text{Tflops}_{\text{Pruned-model}}}{\text{Tflops}_{\text{Baseline}}} \times 100$). Percentage gain = $\frac{\text{Acc}_{\text{Best}} - \text{Acc}_{\text{Baseline}}}{\text{Acc}_{\text{Baseline}}} \times 100$. Best accuracy is highlighted in **bold**; BSBA shows balanced trade-offs.

**Iterative pruning trajectories.** Figure 5 visualizes the iterative layer-pruning process for LLaMA 3.1 8B. Each curve tracks accuracy as layers are progressively removed. As the graphs reveal, the first iteration of TALE typically provides a large boost in accuracy; this boost can make a weak, uncompetitive model competitive. Almost all the trajectories reveal a big initial boost followed by slight increases or decreases; they then follow monotonic decreasing path to accuracies below the baseline and eventually to 0. We stop the iterations once the model accuracy descends below the baseline, and we have found no cases where the trajectory later goes above the baseline. The curve in itself is worthy of future study.

We use this first iteration to guide pruning when trying to balance accuracy with model compression The ⋆ denotes the best-performing pruned model (*Best*), while the □ highlights the *Best Speedup with Baseline Accuracy* (BSBA) model—the pruned configuration achieving maximum compression and inference speedup without falling below the accuracy provided by TALE's first iteration.

From these trajectories, three consistent patterns emerge: (i) TALE identifies compressed models that *outperform* the original across diverse tasks, with ⋆ markers lying strictly above baseline. (ii) Accuracy improvements persist across multiple pruning steps before diminishing returns, showing

that substantial redundancy exists even in carefully tuned pretrained models. (iii) Pruning dynamics are task-specific: datasets such as ARC-Easy and MMLU tolerate deeper pruning while continuing to improve, whereas reasoning-heavy tasks like GSM8K-Hard converge earlier, reflecting heterogeneous layer importance across domains.

**Computation costs** The computational cost of running TALE is modest. For multi-choice tasks such as MMLU, using a validation set of 500 examples, three full TALE iterations complete in $\approx$ 1 GPU-hour on a single A100. Since this pruning is performed once per task, the amortized cost is negligible relative to the inference savings. For details see Appendix C.

**Best vs. BSBA models.** Table 1 compares baseline models against their pruned counterparts under both *Best* and *BSBA* configurations. Across all benchmarks, the Best models yield consistent accuracy gains—up to +146% (LLaMA 8B on GSM8K-Hard), +101% (Lucie 7B on MMLU) and +244% (Qwen 7B on GSM8k-Hard)—while also delivering moderate speedups. BSBA models, by construction, trade smaller gains in accuracy for more aggressive speedups, offering practical operating points when inference cost is the dominant concern.

**Few-shot setting.** We tested TALE under the few-shot regime for Lucie and LLaMA models (Appendix Tables 6–7). Few-shot prompting improves baselines on reasoning tasks such as GSM8K and Math500, yet TALE-pruned variants still achieve higher accuracy in nearly all settings. This shows that pruning-induced improvements are largely complementary to gains from in-context learning.

**Comparisons to other training-free pruning methods**

| Model | Method | Sparsity | WinoGr | HellaSwag | ARC-e | ARC-c |
|---|---|---|---|---|---|---|
| | Baseline | 0% | 69.1 | 76.0 | 74.6 | 46.3 |
| | SpareGPT | 2:4 (50%) | 64.3 | 57.9 | 60.3 | 33.8 |
| | Wanda | 2:4 (50%) | 61.9 | 54.8 | 56.9 | 32.1 |
| LLaMA-2-7B | SliceGPT | 25% | 62.9 | 53.1 | 57.9 | 33.3 |
| | SliceGPT | 30% | 60.8 | 47.9 | 51.4 | 30.9 |
| | SLEB | 10% | 62.4 | 69.3 | 62.7 | 36.9 |
| | TALE | 10% | **73.1** | **80.0** | **76.7** | **54.5** |
| | Baseline | 0% | 72.22 | 79.39 | 77.48 | 49.23 |
| | SpareGPT | 2:4 (50%) | 68.31 | 65.22 | 66.44 | 38.76 |
| | Wanda | 2:4 (50%) | 66.81 | 62.19 | 64.11 | 36.10 |
| LLaMA-2-13B | SliceGPT | 25% | 66.98 | 56.90 | 62.10 | 37.42 |
| | SliceGPT | 30% | 66.11 | 52.39 | 56.12 | 33.17 |
| | SLEB | 10% | 66.93 | 74.36 | 71.84 | 41.55 |
| | TALE | 10% | **76.8** | **83.39** | **80.5** | **53.0** |

Table 2: Accuracies (%) with LM Eval on zero-shot tasks for LLaMA-2-7B and LLaMA-2-13B

| Model | Method | Sparsity | WinoGr | ARC-e | ARC-c |
|---|---|---|---|---|---|
| | Baseline | 0% | 41.2 | 51.7 | 40 |
| LLaMA-2-7B | SLEB | 10% | 18 (-56.3% ↓) | 29 (-43.9% ↓) | 28.8 (-28.0% ↓) |
| | TALE | 10% | **56** (+35.9% ↑) | **62.3** (+20.5% ↑) | **50** (+25.0% ↑) |
| | TALE | 25% | **51** (+23.8% ↑) | **64.8** (+25.3% ↑) | **47.6** (+19.0% ↑) |
| | Baseline | 0% | 42 | 73.0 | 54.9 |
| LLaMA-2-13B | SLEB | 10% | 24.2 (-42.3% ↓) | 43.5 (-40.4% ↓) | 29.8 (-47.3% ↓) |
| | TALE | 10% | **56.4** (+34.3% ↑) | **77.3** (+5.9% ↑) | **64.4** (+17.1% ↑) |
| | TALE | 25% | **55.2** (+31.4% ↑) | **75.3** (+3.2% ↑) | **64.1** (+16.4% ↑) |

Table 3: Accuracies (%) with Our Eval on zero-shot tasks for LLaMA-2-7B and LLaMA-2-13B

Although general training-free pruning techniques often report acceptable accuracy using LM evaluation metrics, they are still far below the accuracy scores gained from TALE (Table 2). Moreover, the accuracy of their decoded outputs deteriorates sharply (3), while TALE increases accuracy on real outputs.

**Takeaways.** TALE consistently uncovers high accuracy and high accuracy/high efficiency models. By balancing task fidelity with computational savings, it enables both accuracy-focused and efficiency-focused deployment. Even for strong models like Qwen 7B we see improvements, and for weaker models like Lucie 7B we see very substantial improvements. Our improvements with

TALE also apply small to language models (Qwen 0.5B). The observed diversity in pruning profiles across datasets underscores the importance of adaptive pruning, rather than one-size-fits-all heuristics, for effective model compression (For a tunable selection metric for choosing among candidate trade-offs see Appendix F). In examining perplexity for pruned models, TALE shows that pruning to optimize for perplexity, though it produces a model with minimal increases in perplexity, does not translate into better performance on downstream tasks, contra Song et al. (2024). In effect perplexity acts as an *another task with its own optimally pruned model*.

## 4.1 TALE AND FINE-TUNING: HOW DOES PRUNING INTERACT WITH FINE-TUNING?

A natural question is whether pruning layers before or after fine-tuning harms the model's ability to learn. One might expect that removing layers reduces representational capacity and thus limits downstream fine-tuning performance compared to baseline instruct-tuned models. Surprisingly, our experiments show the opposite: TALE **not only preserves fine-tuning efficacy but in several cases improves both accuracy and efficiency**.

We explored four settings: (i) fine-tuning the base model (FT), (ii) applying TALE after fine-tuning (FT → TALE ), (iii) pruning first and then fine-tuning (TALE → FT), and (iv) pruning first, then fine-tuning, and finally pruning again (TALE → FT → TALE ). Across various benchmarks, we consistently observed mostly moderate and sometimes significant gains after iterating pruning and fine-tuning, especially on Winogrande and GSM8K (Table 4). This suggests that pruning can act as a regularizer, simplifying the optimization landscape by removing redundant layers.

TALE also reduced computation costs for fine-tuning. For example, pruning LLaMA-3.1 8B before fine-tuning reduced fine-tuning time by 2–2.5 GPU hours on an A100 (an 18.5% reduction) while simultaneously improving Winogrande performance by +2.4%. Iteratively applying pruning and fine-tuning allowed us to prune up to 8 layers achieving still higher accuracy (87.37%) than the full fine-tuned model (85.00%). Similarly, pruning the fully fine-tuned model yielded a 7-layer reduction while maintaining strong accuracy (86.66%).

| Model | Dataset | Baseline | | Pruned Only | | FT Only | | Prune → FT | | FT → Prune | | (Prune → FT) → Prune | |
|---|---|---|---|---|---|---|---|---|---|---|---|---|---|
| | | Perf. | #D | Perf. | #D | Perf. | #D | Perf. | #D | Perf. | #D | Perf. | #D |
| Llama 3.1 8B | Winogrande | 53.83 | 0 | 56.67 | 4 | 85.00 | 0 | 87.06 | 4 | 86.74 | 7 | 87.37 | 8 |
| | MMLU | 54.87 | 0 | 59.90 | 1 | 63.62 | 0 | 63.49 | 1 | 64.21 | 2 | 64.01 | 2 |
| | CommonQA | 72.20 | 0 | 75.30 | 3 | 81.88 | 0 | 81.80 | 3 | 83.40 | 3 | 82.90 | 6 |
| | GSM8K | 15.07 | 0 | 37.08 | 3 | 42.70 | 0 | 53.96 | 1 | 50.86 | 2 | 54.02 | 2 |
| Qwen 0.5B | Winogrande | 49.86 | 0 | 51.88 | 5 | 50.43 | 0 | 50.43 | 5 | 50.49 | 2 | 52.49 | 9 |
| | MMLU | 31.48 | 0 | 39.98 | 2 | 44.87 | 0 | 43.76 | 2 | 45.53 | 2 | 45.58 | 3 |

Table 4: Comparison of **Llama 3.1 8B** and **Qwen 0.5B** across Winogrande, MMLU, and CommonQA under different pruning and fine-tuning regimes. Columns denote: (i) Baseline = original model, (ii) Pruned Only = TALE without fine-tuning, (iii) FT Only = fine-tuned without pruning, (iv) Prune → FT = prune then fine-tune, (v) FT → Prune = fine-tune then prune, (vi) (Prune → FT) → Prune = best fine-tuned-pruned model further pruned. Perf. = performance score, #D = number of deleted layers.

Overall, these results highlight an unexpected but consistent trend: *pruning with* TALE *does not hinder fine-tuning but instead synergizes with it.* Pruning acts like a regularizer, simplifying the optimization landscape, and can effectively interleave with fine-tuning to create models that are both more accurate and computationally efficient. Pruned models fine-tune faster, require fewer parameters to adapt, and are close to or better in performance than their full counterparts.

## 5 INFORMATION THEORY: WHY PRUNED MODELS MIGHT PERFORM BETTER.

Our results pose a puzzle: the increase in accuracy with TALE is counterintuitive: why would removing parts of a carefully trained model lead to better performance? One way to explore this question is mutual information.

Alemi et al. (2016); Tishby & Zaslavsky (2015) use information theory (Shannon, 1948) to analyze how neural networks learn and represent data. Fano & Hawkins (1961) define I(X; Y), the mutual information between two random variables $X$ and $Y$, with the equation:

$$\mathrm{I}(X; Y) = \mathrm{H}(Y) - \mathrm{H}(Y \mid X) = \mathrm{H}(X) - \mathrm{H}(X \mid Y) = \sum_{x \in \mathcal{X}} \sum_{y \in \mathcal{Y}} p(x, y) \log \frac{p(x, y)}{p(x)\, p(y)} \quad (1)$$

where $p(x, y)$ is the joint distribution of X and Y, and $p(x), p(y)$ are their marginals and where $\mathrm{H}(X) = -\sum_x p(x) \log p(x)$ is the Shannon (1948) entropy. I(X; Y) measures how much knowing X reduces uncertainty about Y (Tishby & Zaslavsky, 2015; Shwartz-Ziv & Tishby, 2017). To attempt to explain why accuracy increases through task pruning we also use MI.

A major challenge of this approach is that it requires information about true distributions, which are infeasible to compute. As a result, researchers typically assume a Gaussian distribution Gabrié et al. (2019); Gao et al. (2015); Park et al. (2024) or approximate the probe using a classifier Belinkov (2022); Alain & Bengio (2016) or an MLP Belghazi et al. (2018). These approximations can yield useful insights. In our case, the Gaussian assumption did not fit our datasets. Since we evaluate on QA tasks, we used a trainable classifier to approximate the probes and estimate $I(X^\ell, Y)$ at each layer, where $X^\ell$ denotes the contextualized representations at layer $\ell$ and Y denotes the target answer. This approximates how much information the layer $\ell$ representations contain about the answer. The goal is then to examine whether some layers exhibit a sharp drop in information and whether those layers coincide with the ones whose removal leads to improved performance.

Our findings, summarized in Figure 3 and Table 9, reveal two key patterns: (i) several layers in large pre-trained transformers exhibit a pronounced drop in mutual information; (ii) removing layers dictated by TALE consistently increases the mutual information at the subsequent layer across tasks. Together, these results suggest that certain layers act more as bottlenecks than as contributors to task-relevant representations, providing a rationale for why pruning can lead to improved accuracy.

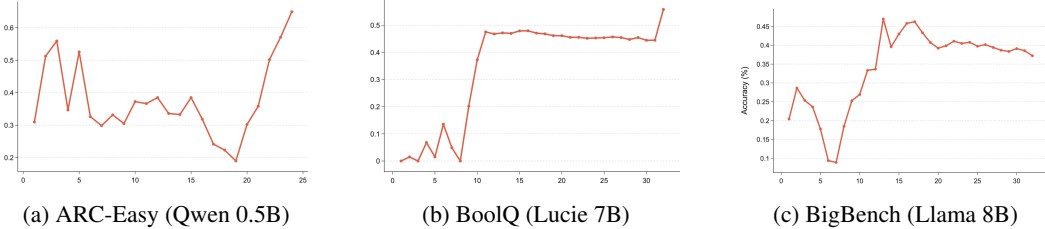

(a) ARC-Easy (Qwen 0.5B)  (b) BoolQ (Lucie 7B)  (c) BigBench (Llama 8B)

Figure 3: Evolution of mutual information (MI) across transformer layers for different benchmark datasets and different models. Each subplot shows how information is processed and transformed as it flows through the network layers, demonstrating distinct patterns of information propagation for (a) ARC-Easy on Qwen 0.5B, (b) BoolQ on Lucie 7B, and (c) BigBench on LLaMA 8B.

## 6 DISCUSSION

We summarize five key observations below from our experiments.

**1. Deleting later layers frequently improves performance on various tasks.** This challenges prior claims that later layers are essential Tenney et al. (2019); Bansal et al. (2023); Song et al. (2025). Even deleting many late layers does not reduce accuracy below baseline, whereas removing even a single early layer is often catastrophic (see Figure 7 in Appendix I). All models exhibit similar behavior. On the other hand, early layers often appear crucial for providing core task-relevant representations that enable the model solve the task, even though probing outputs at those layers does not yield interpretable responses. These results may help model interpretability. Plotting performance degradation from ablating layers helps localize where specific task-solving abilities reside in the network.

**2. Task dependence of layer importance.** Which layers improve or harm performance when removed is highly task dependent. Sometimes a single layer is critical: for instance, removing layer

25 of LlaMA-8B on CommonsenseQA causes a 50-point accuracy drop. Removing LLaMA's layer 3 improves performance on GSM8K-hard but hurts MATH500; the reverse happens when removing layer 11. Removing early layers (1–3) reduces accuracy to near zero on commonsense reasoning tasks (Figure 7), suggesting that certain early layers localize critical task-relevant information. Initial multilingual testing of TALE on Lucie, tuned for French conversational proficiency Gouvert et al. (2025), with bilingual versions of the same data set showed that optimal pruning was task specific rather than language specific.This explains why pruning techniques that remove layers without considering the target task often produce substantial losses in accuracy.

**3. Structured task-specific patterns.** Although pruning is task-specific, related tasks often exhibit similar layer dependencies. Commonsense reasoning tasks (see Figure 7) show importance concentrated in comparable regions of the network. Mathematical reasoning tasks benefit from pruning one to three early layers (e.g., LLaMA layer 3, Mistral layers 6 and 22, Lucie layer 12), but not more (Figures 9, 10, 11). Commonsense and language tasks (ARC, BoolQ, CommonsenseQA, Winogrande, and BIG-Bench) benefit from deleting later layers (Tables 9, 11, 10). This suggests that later layers often play a decoding role for predictions into natural language, which reinforces point 1—pruning them doesn't harm predictive capability.

We observe stronger pruning gains in reasoning-heavy tasks under zero-shot evaluation. All models showed notable accuracy boosts after deleting one or two layers on mathematical reasoning (e.g., LLaMA's and Qwen's triple digit gains on GSM8K-hard, and large gains on for all models on Math500 and GSM8K-hard). By contrast, knowledge-intensive tasks exhibit more modest improvements (e.g., an 11% gain for LLaMA on BIG-Bench).

**4. Model-specific pruning effects.** Different models display distinct pruning behavior. For example, pruned Lucie achieved a 101% gain on MMLU and double-digit gains on ARC-Challenge, CommonsenseQA, BoolQ and GSM8K-hard. While Qwen-7B, LLaMA-8B and Mistral share a similar architecture and scale, they had modest gains on these datasets. Lucie also benefitted from more substantial pruning than the other models. Interestingly, Lucie was trained on a much smaller dataset (3T tokens vs. 15T for LLaMA and 13T for Qwen). This suggests intriguing interactions between pretraining and pruning efficiency. We hypothesize that models trained close to their performance ceiling (via large-scale pretraining, instruction tuning or RLHF) yield smaller pruning gains, whereas models trained under limited objectives may benefit more. But even the Qwen-0.5B trained on a large corpus showed strong pruning efficiency gains (Table 14).

We experimented with producing pruned models for several tasks. We get a LLaMA math model better than baseline LLaMA for both Math500 and GSM8K tasks by dropping layer 12. Taking an intersection of BSBA models for several tasks improved speed up without much loss in accuracy across multiple tasks (Table 15). A better method would be for TALE to prune models on several tasks at once with different mixtures of data to guide the pruning.

## 7 CONCLUSIONS

TALE removes layers irrelevant to a given task $T$ that consistently yields performance above the base model on $T$ and far above the state of the art in pruning without retraining. TALE also reduces computation costs. It can also profitably interact with further training or fine tuning further increasing task specific performance. TALE is a generic strategy and can prune at many levels: base pre-trained models, instruction-tuned models (as we mainly do here), fine-tuned, and post-trained models with RLHF.

TALE can benefit high-throughput applications with time constraints–e.g. in multi-agent systems with task-specific agents or interactive AI assistants. TALE can also help organizations that face critical trade-offs between model capability and computational efficiency use large language models at scale.

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

## A    IMPLEMENTATION DETAILS

**Hardware.**    All experiments were conducted on 1 NVIDIA A100 GPU with 80GB memory.

**Models.**    We applied TALE to five open-weights LLMs of varying scales: **Qwen2.5-0.5B-Instruct**, **Qwen2.5-7B-Instruct**, **Lucie-7B-Instruct**, **Mistral-7B-Instruct**, and **Llama-3.1-8B-Instruct**.

**Datasets for TALE pruning.**    The greedy layer-pruning algorithm was evaluated across nine widely used benchmarks covering reasoning, commonsense, and knowledge-intensive tasks: **ARC-Challenge**, **ARC-Easy**, **MMLU**, **Winogrande**, **GSM8K**, **MATH500**, **CommonQA**, **BIG-Bench**, and **BoolQ**.

**Pruning setup.**    At each iteration, TALE evaluates all candidate single-layer deletions with respect to validation accuracy. The pruning threshold was defined as the baseline accuracy of the full model, ensuring that pruning never reduces performance relative to the original unpruned model. The iterative procedure terminates once no further layer removals satisfy this criterion.

**Fine-tuning setup.**    For fine-tuning experiments, we focused on **Winogrande** and **MMLU**. We employed LoRA with rank 64, a batch size of 4, and the optimizer `paged_adamw_32bit`. A cosine learning rate scheduler was used, and models were trained for 10 epochs.

**Evaluation.**    The LM-Eval methodology presents a significant limitation: it selects the answer with the highest probability among the provided options rather than assessing what the model would actually generate. This approach ignores hallucination behavior and systematically inflates scores; for example, in a two-choice setting, a hallucinated answer still has a 50% chance of being counted as correct. Furthermore, LM-Eval often assigns relatively high scores to weak models, compressing performance differences and making stronger approaches appear only marginally better despite substantial real-world gains. This produces a misleading picture of model capability, as high LM-Eval results do not guarantee that a model will produce correct, coherent outputs in practice. For these reasons, we relied primarily on Our Eval that measures actual accuracy based on the model's generated outputs, which we implemented for each task.

**Prompting.**    For zero-shot and few-shot evaluation, we used task-specific prompts. Below we show the prompt used for datasets, consisting of a system instruction :

---
**ARC-E & ARC-C System Prompt**

You are a Science expert assistant. Your task is to answer multiple-choice science questions at grade-school level. Each question has four answer choices, labeled A, B, C, and D.
For each question: - Carefully read the question and all answer choices. - Select the single best answer from the options (A, B, C, or D). - Respond only with the letter of the correct answer, and nothing else—no explanation or extra words.
Be precise and consistent: Only the answer letter.

---
**Bigbench System Prompt**

"You are a boolean expression evaluator. You must respond with exactly one word: either 'True' or 'False'. Do not provide explanations, steps, or any other text. Only respond with 'True' or 'False'."

---

### BOOLQ System Prompt

"You are a helpful assistant that answers True/False questions based on given passages. Read the passage carefully and determine if the question can be answered as True or False based on the information in the passage. "Respond with only 'A' for True or 'B' for False."

### CommonQA System Prompt

"You are a helpful assistant that answers multiple-choice questions requiring commonsense knowledge and reasoning. Read each question carefully and select the most logical answer from the given options based on common knowledge and reasoning. Respond with only the letter of your chosen answer (A, B, C, D, or E)."

### GSM8K System Prompt

"You are a math problem solver. Solve the given math problem step by step. " "Show your complete reasoning and calculations. " "At the end, write your final answer after '####' like this: #### [your final numerical answer]""

### MMLU System Prompt

"You are a helpful assistant that answers multiple-choice questions across various academic subjects including humanities, social sciences, STEM, and professional fields. Read each question carefully and select the best answer from the given options. Respond with only the letter of your chosen answer (A, B, C, or D)."

### Winogrande System Prompt

You are a careful math problem solver. Show complete step-by-step reasoning and all calculations needed to arrive at the answer. Use clear, numbered or labeled steps so the reasoning is easy to follow.

**IMPORTANT (formatting):**

- After the full reasoning, write the **final answer on a new line by itself** in exactly this format:

  ####
  *integer*

- `<integer>` must be digits only, optionally with a leading "-" for negatives (e.g., $-7$).

- Do **not** add words, punctuation, units, or commentary on the same line as the #### line.

- The #### line must be the **final line of the output** (nothing may follow it).

- Assume all problems expect integer answers; ensure the final line contains a single integer.

## B  NUMBER OF PARAMETERS PER LAYER FOR EACH MODEL

| Model | LLaMA 3.1 8B | Qwen 2.5 7B | Mistral 7B | Lucie 7B | Qwen 2.5 0.5B |
|---|---|---|---|---|---|
| Parameters | 218,112,000 | 233,057,792 | 218,112,000 | 192,946,176 | 14,912,384 |

Table 5: Model parameter counts comparison. LLaMA 3.1 8B, Mistral 7B and Lucie 7B has 32 layers, Qwen 2.5 7B has 28 layers and Qwen 2.5 0.5B has 24 layers.

## C  PRACTICAL COMPUTING SAVINGS AND SCALING

We quantify TALE's inference-cost reduction by measuring TFLOPs (tera-FLOPs) drop per re-moved layer. Across models and tasks, removing a single transformer layer yields a mean TFLOPs reduction of $3.00\% \pm 0.20\%$. Because TALE removes entire layers sequentially, the total TFLOPs reduction scales essentially linearly with the number of iterations (layers removed). In practice this means only a few iterations are required to reach common sparsity targets: e.g., three iterations remove roughly $\approx 9\%$ TFLOPs, sufficient to realize 10% sparsity in our settings.

## D  INTUITION BEHIND TALE

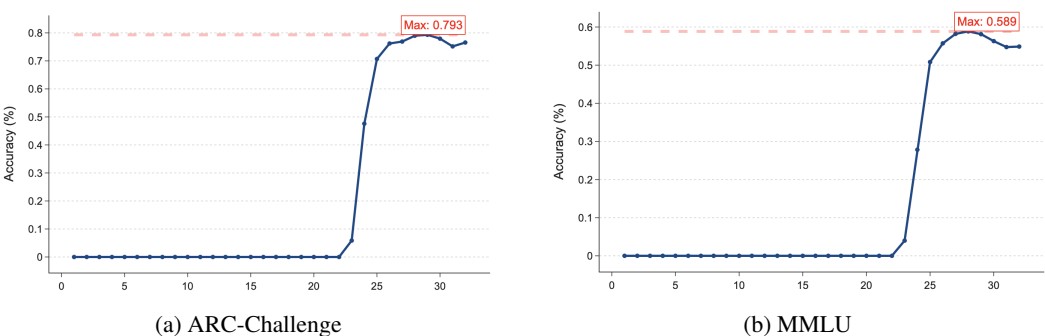

(a) ARC-Challenge        (b) MMLU

Figure 4: Layer-wise output performance for LLaMA models: results when generating predictions from intermediate layers 1 through 32 on three different datasets.

## E  RESULTS

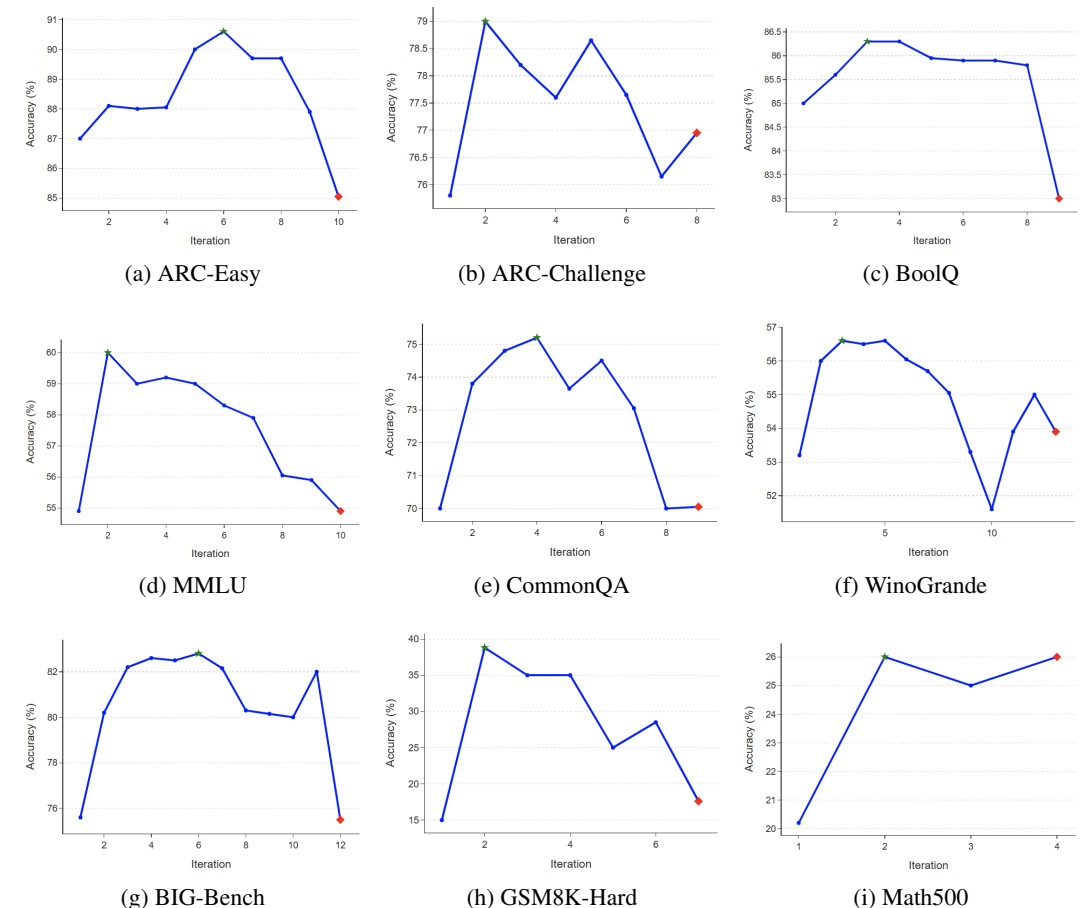

Figure 5: Accuracy progression of TALE across 9 benchmark datasets for LLaMA 3.1 8B. Each curve represents the accuracy at successive iterations. The ★ denotes the best-performing layer drop configuration, while the □ highlights the Best Speed up with at least Baseline Accuracy (BSBA) configuration.

## F   A TUNABLE METRIC FOR FINDING ACCURACY VS. SPEED UP OPTIMIZATION

To systematically select among these candidates according to user priorities, we propose the Accuracy–Efficiency Harmonic Mean (AE-HM):

$$r_A = \frac{\text{Acc(Model)}}{\text{Acc(Baseline)}}, \qquad \text{AE-HM(Model)} = \frac{(1+\lambda^2)r_A\,S}{\lambda^2 S + r_A} = \frac{1+\lambda^2}{\frac{\lambda^2}{r_A} + \frac{1}{S}} \qquad (2)$$

where $S$ denotes the relative inference speedup and $\lambda$ controls the relative importance of accuracy versus efficiency. The user can set AE-HE's parameter $\lambda$ to desired specifications: if $\lambda > 1$, we prioritize $r_A$; if $\lambda < 1$ we prioritize Speedup.

By computing AE-HM for candidate models, we can automatically identify the model with the highest score for a given task or a set of tasks given a particular AE-HM parameter setting:

$$M_{\text{best-compromise}} = \arg\max_i \text{AE-HM}(M_i) \qquad (3)$$

| Dataset | Lucie 7B few-shots | | | | | | |
|---|---|---|---|---|---|---|---|
| | Baseline | Best Model | | | BSBA | | |
| | Perf. | Perf. | #D | Sp. | Perf. | #D | Sp. |
| ARC-Easy | 69.2 | 72.36 | 9 | 1.41 | 71.27 | 12 | 1.68 |
| ARC-Challenge | 49.31 | 55,17 | 9 | 1.39 | 51.72 | 13 | 1.67 |
| BoolQ | 77.6 | 79.10 | 6 | 1.22 | 78.5 | 10 | 1.27 |
| MMLU | 41.02 | 43.44 | 7 | 1.26 | 41.48 | 11 | 1.55 |
| COMMONQA | 55.4 | 69.7 | 3 | 1.22 | 57.10 | 17 | 2.02 |
| WINOGRANDE | 52.8 | 56.90 | 12 | 1.58 | 53.30 | 17 | 1.74 |
| BIG-Bench | 68.8 | 77.20 | 9 | 1.61 | 72 | 15 | 2.23 |
| GSM8K-HARD | 26.97 | 29.21 | 1 | 1.03 | 26.97 | 2 | 1.1 |

Table 6: Results of **Lucie 7B** across nine benchmarks. All tested on 5-shots, except gms8k on 8-shots Performance (%) cells are color-coded: green = gain, red = decline, and gray = near-neutral change compared to baseline.

| Dataset | LLaMA 3.1 8B few-shots | | | | | | |
|---|---|---|---|---|---|---|---|
| | Baseline | Best Model | | | BSBA | | |
| | Perf. | Perf. | #D | Sp. | Perf. | #D | Sp. |
| ARC-Easy | 90.36 | **92.18** 2.01% ↑ | 4 | 1.14 | 90.91 | 8 | 1.37 |
| ARC-Challenge | 78.2 | **83.10** 6.27% ↑ | 3 | 1.17 | 78.62 | 9 | 1.42 |
| BoolQ | 82.7 | 85.3 3.1% ↑ | 4 | 1.11 | 83.0 | 6 | 1.22 |
| MMLU | 59.2 | 62.38 5.37% ↑ | 4 | 1.14 | 59.57 | 7 | 1.26 |
| COMMONQA | 73.30 | 75.30 2.72% ↑ | 6 | 1.22 | 73.80 | 7 | 1.32 |
| WINOGRANDE | 57.01 | 60.15 5,26% ↑ | 3 | 1.1 | 57.02 | 8 | 1.3 |
| BIG-Bench | 70.0 | 83.60 19,43% ↑ | 5 | 1.2 | 81.20 | 15 | 1.83 |
| GSM8K-HARD | 60.67 | 60.67 | 0 | 1 | 60.67 | 0 | 1 |
| MATH500 | 44.00 | 49.00 11.36% ↑ | 1 | 1.02 | 45.00 | 2 | 1.03 |

Table 7: Results of **LLaMA 3.1 8B** across nine benchmarks. All tested on 5-shots, except gms8k and MATH500 on 8-shots

## G DELETED LAYERS IN EACH MODEL AND BENCHMARK

| Dataset | Best Model | BSBA |
|---|---|---|
| ARC-Easy | 19  25  27  28 | 19  20  21  24  25  26  27  28 |
| ARC-Challenge | 19  22  27 | 19  20  21  22  23  24  26  27  28 |
| BoolQ | 19  25  26  32 | 15  19  21  22  25  26  30  32 |
| MMLU | 20  21  27  28 | 20  21  22  24  27  28  32 |
| CommonQA | 21  22  27  28  31  32 | 21  22  23  27  28  31  32 |
| Winogrande | 20  22  24 | 17  19  20  22  24  26  29  32 |
| BIG-Bench | 11  16  20  21  26 | 10  11  16  20  21  22  23  24  26  27  28  29  30  31  32 |
| MATH500 | 28 | 24  28 |

Table 8: Deleted layers represented as color-coded inline numbers. Blue = Best Model, Orange = BSBA for LlaMA 3.1 8B with few-shots.

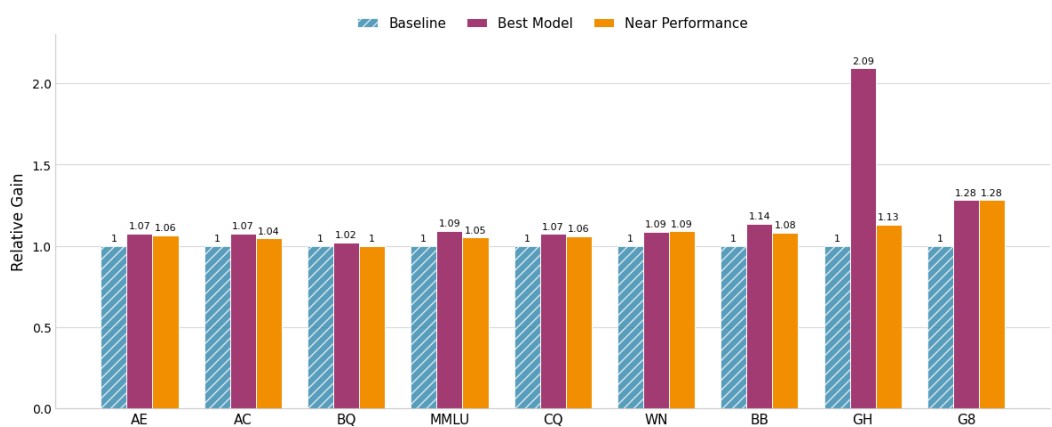

Figure 6: Relative Gain comparison across datasets. LLaMA $\beta = 3$

Table 9 shows how using AE-HM allows us to bring model size down effectively on our BSBA Llama model with 0 shot performance on our nine data sets. The BSBA LLama model had speed up gains between 27 and 46% on our various benchmarks and maintained performance at or above original model levels (See Table 9).

| Dataset | Best Model | BSBA |
|---|---|---|
| ARC-Easy | 19 20 21 29 32 | 19 20 21 22 25 27 29 32 |
| ARC-Challenge | 19 20 23 27 | 19 20 21 23 25 27 28 |
| BoolQ | 21 23 28 | 18 21 22 27 28 32 |
| MMLU | 21 | 19 21 22 24 25 26 27 28 31 |
| CommonQA | 19 23 28 | 19 22 23 26 27 28 |
| Winogrande | 23 24 26 32 | 20 21 22 23 24 25 26 27 29 31 32 |
| BIG-Bench | 14 20 22 28 29 | 14 18 20 21 22 23 24 28 29 31 32 |
| GSM8K-Hard | 3 | 3 21 22 25 26 27 29 |

Table 9: Deleted layers represented as color-coded inline numbers. Blue = Best Model, Orange = BSBA for LlaMA 3.1 8B 0 shot.

| Dataset | Best Model | BSBA |
|---|---|---|
| ARC-Easy | 19 22 28 | 6 19 22 24 26 27 28 |
| ARC-Challenge | 27 28 | 7 22 23 26 27 28 |
| BoolQ | 18 21 27 28 | 12 19 21 22 26 27 28 |
| MMLU | 22 23 26 27 28 | 18 22 23 26 27 28 |
| CommonQA | 22 28 | 6 21 22 23 27 28 |
| Winogrande | 22 26 27 | 6 20 22 25 26 27 |
| BIG-Bench | 10 19 23 25 26 27 | 10 19 23 25 26 27 |

Table 10: Deleted layers represented as color-coded inline numbers. Blue = Best Model, Orange = BSBA for **Qwen 2.5 7B** zero-shot.

| Dataset | Best Model | BSBA |
|---|---|---|
| ARC-Easy | 15 16 23 24 27 28 | 13 15 16 18 19 20 21 22 23 24 25 27 28 |
| ARC-Challenge | 16 18 20 21 23 25 26 | 15 16 18 19 20 21 22 23 25 26 28 |
| BoolQ | 8 17 25 28 29 | 5 8 11 12 13 14 15 16 17 19 20 23 25 26 27 28 29 31 |
| MMLU | 11 12 15 16 20 21 22 28 | 5 10 11 12 13 14 15 16 17 18 19 20 21 22 23 24 25 26 28 30 31 |
| CommonQA | 11 12 27 | 11 12 13 15 16 17 18 19 20 21 22 23 24 25 27 28 |
| BIG-Bench | 6 7 15 17 20 21 25 26 27 | 6 7 13 15 17 19 20 21 22 24 25 26 27 28 29 |
| GSM8K-Hard | 12 | 12 21 23 |

Table 11: Deleted layers represented as color-coded inline numbers. Blue = Best Model, Orange = BSBA for Lucie 7B 0 shots.

| Dataset | Best Model | BSBA |
|---|---|---|
| ARC-Easy | 21 22 24 26 29 | 21 22 23 24 25 26 29 30 32 |
| ARC-Challenge | 22 24 25 27 28 30 | 21 22 24 25 26 27 28 30 |
| BoolQ | 17 22 23 24 27 32 | 12 17 21 23 24 25 27 28 32 |
| MMLU | 24 30 | 22 23 24 25 26 27 30 32 |
| CommonQA | 19 22 25 28 | 19 21 22 24 25 28 32 |
| Winogrande | 18 19 20 22 23 24 26 27 31 32 | 4 13 18 19 20 22 23 24 26 27 29 31 32 |
| BIG-Bench | 3 5 15 22 23 24 26 27 28 | 3 5 14 15 18 22 23 24 26 27 28 |
| GSM8K-Hard | 6 22 | 6 11 22 28 |

Table 12: Deleted layers represented as color-ccdinline numbers. Blue = Best Model, Orange = BSBA for **Mistral** zero-shot.

# H ABLATION STUDY ON VALIDATION SET OF PRUNING

We analyze the effect of validation set size on TALE's layer selection. Table 13 reports the specific layers dropped for different validation set sizes across three tasks (ARC-Easy, MMLU, GSM8K) and two models (Llama 3.1 8B, Qwen 2.5 7B).

| Model | Val Size | Task | Dropped Layers |
|---|---|---|---|
| **Llama 3.1 8B** | 200 | ARC-E | {19, 20, 22, 29, 32 } |
| | | MMLU | { 21 } |
| | | GSM8K | { 3 } |
| | 500 | ARC-E | { 19, 20, 21, 29, 32 } |
| | | MMLU | { 21 } |
| | | GSM8K | { 3 } |
| | 1000 | ARC-E | { 19, 20, 21, 29, 32 } |
| | | MMLU | { 21 } |
| | | GSM8K | { 3 } |
| **Qwen 2.5 7B** | 100 | ARC-E | { 22 , 27 , 28 } |
| | | MMLU | { 18 , 22 , 24 , 27 , 28 } |
| | | GSM8K | { 19 } |
| | 500 | ARC-E | { 19 , 22 , 28 } |
| | | MMLU | { 22 , 23 , 26 , 27 , 28 } |
| | | GSM8K | { 19 } |
| | 1000 | ARC-E | { 19 , 22 , 28 } |
| | | MMLU | { 22 , 23 , 26 , 27 , 28 } |
| | | GSM8K | { 19 } |

Table 13: Layers removed by TALE for different validation-set sizes across three tasks. This reveals the stability of pruning decisions directly.

# I   MORE ON PRUNING AND A COMMON PRUNED LAYERS MODEL

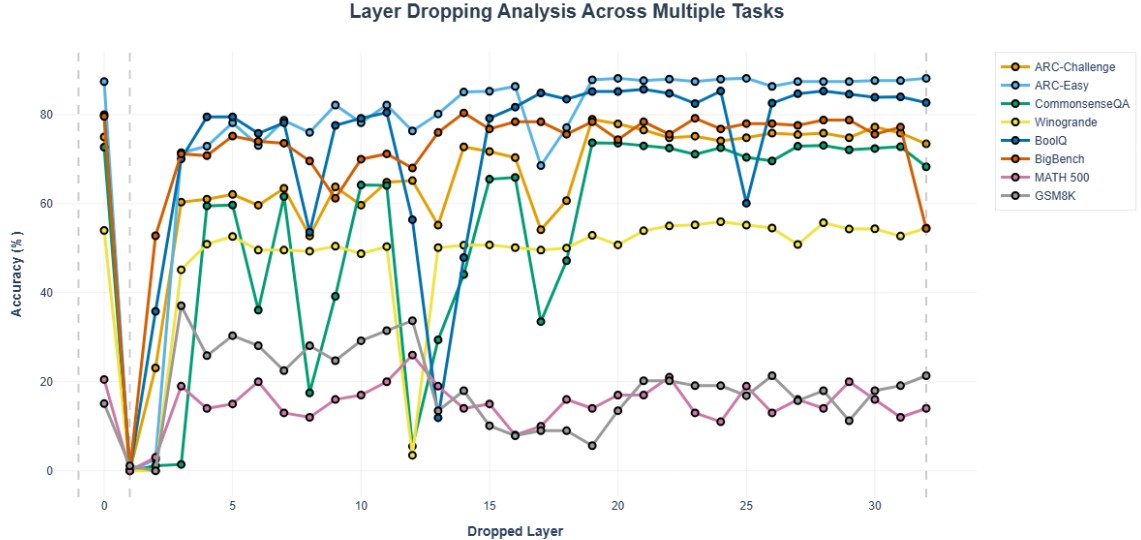

Figure 7: Nine benchmark tasks indicating performance after one layer is dropped from different positions in Llama3-8B.

Table 14: Performance comparison under 0-shot evaluation. Accuracy (**Perf.**) uses Our Eval We also report number of dropped layers (**#D**), and relative inference speedup (**Sp.**) in terms of percentage of Tflops saved (Percentage saved = $\frac{\text{Tflops}_{\text{Baseline}} - \text{Tflops}_{\text{Pruned-model}}}{\text{Tflops}_{\text{Baseline}}} \times 100$). Percentage gain = $\frac{\text{Acc}_{\text{Best}} - \text{Acc}_{\text{Baseline}}}{\text{Acc}_{\text{Baseline}}} \times 100$. Best accuracy is highlighted in **bold**; BSBA shows balanced trade-offs.

| Dataset | Qwen 2.5 0.5B (zero-shot) | | | | | | |
|---|---|---|---|---|---|---|---|
| | Baseline | Best Model | | | BSBA | | |
| | Perf. | Perf. | #D | Sp. | Perf. | #D | Sp. |
| ARC-Easy | 40.00 | **60.91**(+48.49% ↑) | 3 | -9.3% | 48.36 | 5 | -15.5% |
| ARC-Challenge | 35.52 | **40.34**(+13.57% ↑) | 1 | -3.1% | 37.24 | 4 | -12.4% |
| BoolQ | 62.30 | **67.20**(+7.87% ↑) | 5 | -15.5% | 66.20 | 6 | -18.6% |
| MMLU | 31.48 | **39.97**(+26.96% ↑) | 2 | -6.2% | 33.90 | 5 | -15.5% |
| CommonQA | 42.40 | **49.10**(+15.80% ↑) | 2 | -6.2% | 44.00 | 3 | -9.3% |
| Winogrande | 49.86 | **51.88**(+4.51% ↑) | 5 | -15.5% | 49.87 | 17 | -52.6% |
| BIG-Bench | 72.40 | **73.60**(+1.66% ↑) | 2 | -6.2% | 73.60 | 2 | -6.2% |
| GSM8K-HARD | 6.74 | **11.24**(+66.77% ↑) | 1 | -3.1% | 8.99 | 2 | -6.2% |
| Math500 | 8.00 | **12.00**(+50% ↑) | 1 | -3.1% | 9 | 2 | -6.2% |

## J  GENERAL PRUNING RESULTS

| Group | Dataset | Baseline | Pruned Model | speedup |
|---|---|---|---|---|
| Common-sense | ARC-Easy | 87.0 | 87.82 | 1.2 |
| | ARC-Challenge | 75.86 | 75.00 | 1.21 |
| | CommonQA | 72.20 | 64.70 | 1.1 |
| | Winogrande | 54.20 | 50.57 | 1.13 |
| Reading | BoolQ | 85.0 | 85.5 | 1.17 |
| | BIG-Bench | 75.2 | 67.2 | 1.1 |

Table 15: Accuracy of LLaMA-3.1-8B (baseline) versus a pruned variant obtained by dropping layers selected through BSBA. For each task, BSBA identified removable layers, and we retained the intersection of layers that appeared in at least 75% of tasks within the Common-sense group (layers 19, 22, 23, 27) and (layers 18, 21, 22, 28, 32) for Reading Comprehension tasks. These layers were then pruned globally from the model, and performance was re-evaluated across tasks. Speedup is reported relative to the baseline.

## K  TALE EVALUATION WITH PERPLEXITY

| Model | WikiText2 | | LAMBADA | |
|---|---|---|---|---|
| | Vanilla | TALE | Vanilla | TALE |
| LLaMA 3.1 8B | 24.6 | 24.9 | 28.1 | 28.9 |
| Lucie 7B | 46.1 | 36.4 | 52.5 | 43.8 |

Table 16: Perplexity scores for two models across WikiText2 and LAMBADA with Vanilla and TALE (sparisty 10%) configurations.

## L  TALE , OUR GREEDY-SELECTION ALGORITHM

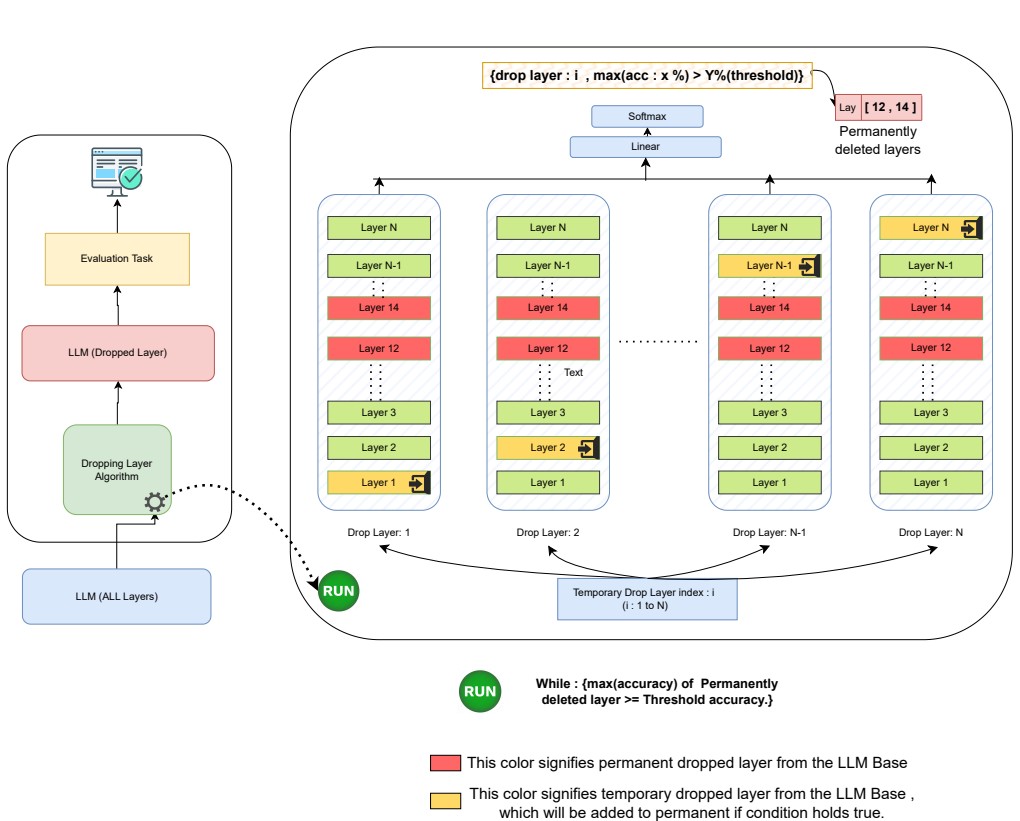

Figure 8: Illustration of TALE layer elimination. Candidate layers (yellow) are tested for removal, and the best-performing ones above the threshold are permanently dropped (red) until no further improvement is possible.

