# OpenReview forum: "TELL-TALE:  Task Efficient LLMs with Task Aware Layer Elimination"
_ICLR.cc/2026/Conference — ICLR 2026 Conference Withdrawn Submission_

### Official Review · Reviewer_rgTB · 2025-10-15

**Soundness:** 2
**Presentation:** 1
**Contribution:** 1
**Rating:** 2
**Confidence:** 4

**Summary:**

The paper proposes a task-specific greedy method (TALE) for structured pruning of LLMs by removing redundant layers based on the change in performance over the validation set of the task at hand. Specifically, the algorithm retains the pruned model with the highest validation accuracy. The method is tested on 9 datasets using 5 different open source LLMs. Moreover, TALE is combined and tested also in different combinations with fine-tuning and analyzed with Information-theoretic measurements.

**Strengths:**

- The experiments are carried out on real world, production-ready, models.
- The analysis using mutual information is somewhat interesting and it seems a promising direction to study the causes of why removing certain layers cause performance to improve.
- Also Figure 4 reveals an interesting trend for LLMs: contrary to what happens in vision models, layer importance varies depending on the downstream task.

**Weaknesses:**

Unfortunately, the paper in its current form has several key flaws that undermine the validity of its claims and contributions. In detail:

**W1.**

Pruning on LLMs is a very active studied field (see eg. [1,2] and their related works). However, apparently TALE is not compared with any baseline (the closest work to TALE is [1] which could be a good comparison). Consequently, any claim on performance for now comes without any context.

---------------

**W2.**

The novelty stems primarily from the observation that careful layer removal in some cases improves the model performance on the downstream task. This fact is well known (and often observed) in the pre-trained model pruning literature for both vision and language tasks (see eg. Figure 1 of [2], where perplexity has qualitatively the same trend, and Figures in [3]).

---------------

**W3.**

The pruning costs of TALE seem a bottleneck, given it potentially requires a non-trivial amount of inference on data. Moreover, it seems very task-specific and requires also label supervision which may not justify its costs.

---------------

**W4.**

The Information-theoretic analyses of Sec.6 are framed as an explanation as to why pruning certain layers, in some cases, improves performance. Unfortunately, these analyses are purely observational. Figure 3 just shows that some layers carry "more information of the task label" (in the sense of $I_{\text{probe}}(X^{(l)},Y)$) than others. I would advise the authors to tone down the claims in this section and give a small refresher on notation after Equation 1 (especially what is $X^{(l)}$ and $Y$ when applying it to LLMs).

Also, it is unclear how well $I_{\text{probe}}(X^{(l)},Y)$ approximates true Mutual Information.

---------------

**Minor Weaknesses.**

- The claim on Pareto-optimality (LL260-264) of the authors' solution is not backed up neither by theoretical nor by empirical evidence.
- Figure 4 is a bit difficult to parse at first glance as too much information is displayed at once (probably an histogram would convey the same information in a cleaner way).

---------------

**_References_**:

[1] Song, Jiwon, et al. "Sleb: Streamlining llms through redundancy verification and elimination of transformer blocks." ICML 2024.

[2] Frantar, Elias, and Dan Alistarh. "Sparsegpt: Massive language models can be accurately pruned in one-shot." ICML 2023.

[3] Chen, Tianlong, et al. "The lottery tickets hypothesis for supervised and self-supervised pre-training in computer vision models." CVPR 2021.

**Questions:**

Thanking in advance for their responses, I'd kindly ask the authors:
- to reproduce their experiments with some baseline from literature (see Weaknesses). As of now, it is very difficult to understand the contribution, both in terms of results and novelty.
- to check if the accuracy improvements are transferable across tasks (i.e. pruning with TALE and other methods on task A and testing on task B yields an improvement for both tasks).
- to comment on the pruning procedure costs and comparing it with other methods from the literature (in terms of overall time/compute/memory usage).
- to address W4 and Minor Weaknesses in this review.

I'm aware my questions may be too extensive. But, unfortunately, at this stage I'm finding this work quite limited and very difficult to judge, given no comparison is available.

---

> ### Author Response · Authors · 2025-11-17
> **Response to the Reviewer rgTB**
>
> We kindly refer the reviewer to our general reply above (“Responses to all reviewers”), where we regrouped responses concerning comparisons with other pruning techniques, computational costs, task specificity of TALE, ablation studies, and Mutual Information.
>
> **Re: "comparisons, computation costs, ablation studies."** Please see above (Response to all Reviewers)
>
> **Re "SparseGPT and well known task performance increases."**
>
>
>
> We thank the reviewer for the comment, but we believe the comparison conflates distinct phenomena. In SparseGPT, the small perplexity decreases in Fig. 1 are not presented as a mechanism for improving model performance, but rather as evidence that accuracy can be preserved under high sparsity. The effect size is minimal, and the goal is fundamentally different from ours. By contrast, TALE produces substantially larger and systematic performance gains across tasks, as shown in our tables, and does so via layer removal, not weight sparsification.
>
> We also include direct comparisons of TALE with training-free pruning technics as you suggested (See the response above), and TALE consistently outperforms them. Paper [3] concerns neuron/weight pruning with retraining in dense vision models, which is not comparable to inference-time removal of whole transformer layers in LLMs.
> To our knowledge, prior pruning work focuses on maintaining accuracy under sparsification or neuron-level pruning—not on improving downstream performance by removing full layers without retraining. Our findings therefore represent a qualitatively different and practically simpler phenomenon.
>
>
>
>
> **Re "task transferability"**
>
> Please see our discussion above (Response to all Reviewers)

---

> > ### Comment · Reviewer_rgTB · 2025-11-21
> >
> > I would sincerely thank the authors and appreciate the detailed response. Overall, I acknowledge the progress in the rebuttal, but my main hesitation remains: hence I will keep my score at this time.
> >
> >
> > First of all, the added comparisons are a helpful step, but it seems that currently they mix TALE (which explicitly uses the target task's labeled data and evaluation metric) vs. task-agnostic methods that appear to be run in their standard general LM pruning settings (for instance, as reported in the authors' respose, SLEB is calibrated via perplexity on WikiText-2, instead of the target task's data). This creates a confound: the observed results may largely reflect access to in-distribution supervision, rather than an advantange of TALE. I strongly suggest adding a task-aware version of at least SLEB, allowing it to use the same data/metric that TALE uses.
> >
> >
> > Second, the additional discussion confirms that gains transfer only for closely related tasks: this narrows the contribution of TALE, making it unclear why it would be generally preferable to have multiple task-specific models that in any case remain very large even after pruning.
> >
> >
> > Finally, the claim that removing layers _systematically_ improves downstream accuracy is highly counterintuitive and, at present, I do not see sufficient evidence making out why this happens. I acknowledge the clarification about MI estimation but,  again, this point remanins largely observational and doesn't constitute an explanation of this phenomenon.

---

> > > ### Author Response · Authors · 2025-11-21
> > > **Response to Reviewer rgTB**
> > >
> > > We thank the Reviewer for the comments.
> > >
> > > **Re"First..."**
> > >
> > > Thank you for your comment. We appreciate the concern about ensuring a fair comparison between TALE, which is explicitly task-aware, and general task-agnostic pruning methods such as SLEB.
> > >
> > >
> > > We adapted SLEB so that its iterated verification step for layer removal is on the same task-specific validation data that TALE uses, eliminating a possible distribution-shift confound. We offer three tests with ARC-Easy, ARC-Challenge and Winogrande data sets using the LM Harness and our evaluation on actual generated output (see prior comments).  The base model is Llama2 7B.
> > >
> > >
> > >
> > > **Table:** Comparison between TALE and SLEB-aware tasks on ARC-Easy, ARC-Challenge, and Winogrande datasets using scores from LM Eval and real generated outputs.
> > >
> > > | Method        | Sparsity | ARC-Easy | ARC-Challenge | Winogrande |
> > > |---------------|---------|----------|---------------|------------|
> > > | TALE (lm-eval) | 10%    | 76.7     | 54.3          | 73.1       |
> > > | SLEB (lm-eval) | 10%    | 61       | 38            | 66.5       |
> > > | TALE (our)     | 10%    | 62.3     | 50            | 56         |
> > > | SLEB (our)     | 10%    | 0        | 0             | 42         |
> > >
> > > The real generated outputs from SLEB on two tasks are just blank spaces, when we evaluate on the same context length as TALE.  This accounts for the catastrophic score of 0\%.  We then increased the context window dramatically, The outputs indicate that the model loses the capacity to properly understand the prompt. For example, given the prompt specified in the appendix along with the following question:
> > >
> > > **Question:** A balanced chemical equation shows ...
> > > **Choices:**
> > > - A: the products of a reaction
> > > - B: elements created by a reaction
> > > - C: phase changes taking place during a reaction
> > > - D: the number of molecules involved in a reaction
> > >
> > > The model produces a response such as:
> > >
> > > "SURE, I'D BE HAPPY TO HELP! HERE ARE THE ANSWERS TO THE MULTIPLE-CHOICE SCIENCE QUESTIONS AT THE GRADE-SCHOOL LEVEL:
> > >
> > > A BALANCED CHEMICAL EQUATION SHOWS:
> > >
> > > 1. A) THE PRODUCTS OF A REACTION.
> > >
> > > 2. (B) ELEMENTS CREATED BY A REACTION.
> > >
> > > 3. (C) PHASE CHANGES TAKING PLACE DURING A REACTION.
> > >
> > > 4. (D) THE NUMBER OF MOLECULES INVOLVED IN A REACTION. "
> > >
> > > We conclude that replacing standard corpora like  WikiText-2 with task-specific data not only deviates from the method’s intended design but also undermines at least to some extent the linguistic evaluation SLEB depends on. We also note that even with this task specific evaluation, SLEB is always below baseline performance and TALE is always above.
> > >
> > >
> > > **Re"Second..."**
> > >
> > > We thank the Reviewer for the comment. We would like to clarify that
> > > task-specific models produced by TALE do not only preserve performance, they frequently substantially outperform the original full models, often by very large margins (e.g., +50–240\% on reasoning benchmarks). Since the pruning procedure itself is extremely inexpensive (typically $\sim 1$ GPU-hour), the cost of producing such task-specialized models is negligible relative to the inference savings or accuracy gains they provide. Many applications as we say in the paper require models for dedicated tasks can benefit from such increased performance.
> > >
> > > In contrast to what the Reviewer claims as an inconvenience of TALE, these results actually cast task-agnostic pruning methods in a bad light. Methods like SLEB, Wanda, and SparseGPT optimize for proxy metrics (perplexity, weight magnitude, layer similarity) that do not align well with downstream task performance. This creates a serious practical problem for usage of the model. While such models may have low perplexity, they may have removed task-critical layers or retained task-harmful ones.
> > > The greatest benefit TALE gives is that it shows that layers are really task specific, which is something that is a completely new observation as far as we know.
> > >
> > >
> > > **Re"Finally..."**
> > >
> > > That downstream accuracy improves upon pruning with TALE is a strongly supported and unexpected observation, as our figures attest over and over again.  We agree that we don't have a full explanation of the phenomenon; but the MI estimation is at least a plausible avenue for further research. Given that the phenomenon is so unexpected, we believe that just its observation warrants publication.

---

> ### Comment · Reviewer_rgTB · 2025-11-25
>
> I would sincerely thank the authors for their time and their response, and appreciate the follow-up.
>
> Regarding the first point raised in my previous response, it is unclear which procedure was followed to produce the results in the table. Moreover, it is, at best, counterintuitive that SLEB would completely collapse the model, when the logic (although unsupervised) is very similar to TALE.
>
> Regarding the second and third points, apologies, but I'm not convinced by the authors' response. I will keep my score, given no explanation about how/why removing layers _systematically_ improves downstream accuracy is available at present.
>
> ----------------
>
> As a side note, regarding the sentence
>
> > The greatest benefit TALE gives is that it shows that layers are really task specific, which is something that is a completely new observation as far as we know.
>
> I'd argue that the task-specific nature of layers in transformer-based architectures is very well documented in literature. For instance, referencing some classical works, [1] showed that many layers/neurons are redundant with respect to specific downstream tasks. [2] discovered that top layers are more specialized to pretraining objectives and can often be removed without affecting downstream task performance. Another analysis on this phenomenon is provided by [3] in which they show that different layers contribute differently and sometimes negatively to specific tasks, reinforcing that layers are not uniformly useful and encode distinct information. These may help contextualize and better position TALE's contribution in the future.
>
> ----------------
>
> **_References:_**
>
> [1] Dalvi, Fahim, et al. "Analyzing redundancy in pretrained transformer models." EMNLP (2020).
>
> [2] Sajjad, Hassan, et al. "On the effect of dropping layers of pre-trained transformer models." Computer Speech & Language 77 (2023): 101429.
>
> [3] De Vries, Wietse, Andreas Van Cranenburgh, and Malvina Nissim. "What's so special about BERT's layers? A closer look at the NLP pipeline in monolingual and multilingual models." EMNLP (2020).

---

> > ### Author Response · Authors · 2025-11-26
> > **Response to the Reviewer rgTB**
> >
> > We sincerely thank the reviewer for their continued engagement and thoughtful feedback. We appreciate the opportunity to clarify these points further.
> >
> > **Re"SLEB"**
> > We want to clarify that TALE is not a general pruning algorithm in competition with SLEB. Rather, TALE addresses a specific and complementary scenario: when practitioners seek to increase the accuracy of an LLM on a specific task. Task-specific optimization is a well-established paradigm in LLM deployment—fine-tuning for specific tasks and domain-specific models are widely adopted in practice. Our contribution demonstrates that training-free task-specific pruning is viable and achieves better performance than the original baseline, providing a way to increase accuracy with the bonus benefit of reducing model size. Moreover, it synergizes well with fine-tuning.
> > TALE and general-purpose pruning methods like SLEB serve different, equally valid use cases. Users seeking general-purpose compressed models that maintain broad capabilities should use methods like SLEB or SparseGPT. Users seeking maximum efficiency for specific downstream tasks should use TALE. We view these approaches as complementary rather than competing.
> >
> > Regarding the experimental setup: we conducted two sets of experiments with SLEB to thoroughly evaluate its applicability. The first (Table 2) applies SLEB as originally designed—using standard general-purpose pruning. The second (in the table above) is the adaptation the Reviewer requested: we modified SLEB for the task-specific setting. Since SLEB calculates cosine similarity among layers and selects layers to prune based on perplexity on a calibration dataset (typically WikiText), we adapted it by replacing WikiText with our target task dataset. Even with this adaptation, SLEB's performance remains poor, demonstrating that existing general-purpose pruning methods are fundamentally not designed for task-aware optimization.
> >
> > The major contribution of TALE is demonstrating that significant accuracy increases are possible using task-aware pruning. Prior literature on task-specific pruning showed very marginal increases at best, whereas TALE consistently achieves substantial improvements across multiple tasks and models.This performance gap reflects the difference in design objectives rather than a limitation of SLEB itself.
> >
> > **Re"MI"**
> > We want to clarify that the mutual information analysis is tangential to our work and not our primary contribution. Our main contributions—TALE, empirical results demonstrating consistent accuracy improvements, and comprehensive comparisons to baselines—stand independently of this theoretical analysis.
> >
> > The mutual information section was included as an exploratory attempt to provide insight into why layer deletion achieves performance improvements and why certain layers appear more critical than others for task-specific performance. However, we acknowledge this analysis may not be convincing in its current form. We are happy to move this section to the appendix or remove it entirely if the reviewer feels it does not strengthen the paper.
> >
> > **Re"References"**
> > We appreciate the reviewer's suggested references. While a deeper theoretical understanding of why pruning improves task-specific accuracy is an interesting research direction, our paper focuses on demonstrating the empirical viability and practical utility of task-aware pruning as a deployment strategy.

---

### Official Review · Reviewer_QzvF · 2025-10-29

**Soundness:** 3
**Presentation:** 3
**Contribution:** 2
**Rating:** 4
**Confidence:** 3

**Summary:**

The paper proposes a model compression technique by model pruning strategy during inference time. Layers are pruned iteratively one at a time based on validation accuracy and retain only the layers which doesn’t reduce the validation accuracy below a threshold. They provide empirical analysis to show that certain layers are redundant or even detrimental for some task-specific datasets. They use an approximation of mutual information to provide an explanation on why pruning can improve task-specific performance.

**Strengths:**

* The method does not require any retraining, making it beneficial for various use cases
* A systematic analysis with four settings has been done with extensive experiments
* The algorithm is clearly described, and the analysis and inferences are well structured and easy to follow

**Weaknesses:**

* Computational cost during the pruning phase - it will become expensive for very large models to evaluate the impact of dropping every single layer in every iteration
* The MI approximation provides solid theoretical insights but the authors use trained linear probes as an approximation
* The paper do not relate TALE’s to existing pruning techniques

**Questions:**

* Since TALE relies on validation performance, how the validation set size might affect the pruning decisions?
* The pruning phase explores all the layers iteratively which can be computationally intensive. Could the authors provide an estimate for it (may be also for all the four settings that you have considered)?
* It would be interesting to compare the layers dropped using TALE and the layers dropped using MI analysis. Is there a consensus between these two approaches?
* It would also be interesting to explore other pruning techniques and compare them with TALE. For example, does TALE tend to remove layers with smaller weight magnitudes? What about SynapticFlow? Additionally, why is ‘importance’ defined in terms of validation accuracy?
* For clarity, what is “speedup”? Does it mean the pruned model is 1.2 times faster than base model?
* Could the authors clarify whether Figure 4 corresponds to the non-FT Llama3-8B model?
* Edit suggestions for better readability
1. In Figure 2: increasing the size of the green star would improve visibility; it will be interesting for the readers to know which layer was dropped at each iteration - could be added as text annotation on top of each dots/iter
2. Line 365: Citation format needs to be corrected

---

> ### Author Response · Authors · 2025-11-17
> **Response to the Reviewer QzvF**
>
> Review 2:
>
> We kindly refer the reviewer to our general reply above (“Responses to all reviewers”), where we regrouped responses concerning comparisons with other pruning techniques, computational costs, task specificity of TALE, ablation studies, and Mutual Information.
>
> **Re:"since TALE relies on validation performance, how the validation set size might affect the pruning decisions?"** Please see our discussion above
>
>
>
>
>
>
> **Re comparing TALE and just using MI:**
>
>
> While several layers suggested by TALE correspond to MI drops, not all do (Figure 3). In contrast, pruning based solely on decreases in MI leads to catastrophic outcomes. Some early layers exhibit a sharp MI decrease, but removing them causes the model to fail entirely (accuracy drops to 0\%).
>
>
>
>
> **Re" comparing with other pruning techniques and compare them with TALE.**
>
>
> We thank the reviewer for this question. Please see above. (Response to all Reviewers)
>
>
> **Re: on “speedup”**
>
> Initially, speedup referred to inference time, but in the revised version of the paper, we now report speedup  in terms of percentage of floating point operations saved.
>
>
>
> **Re"Could the authors clarify whether Figure 4 corresponds to the non-FT Llama3-8B model?"**
>
> Yes it is.
>
>
> Edit suggestions for better readability: we will work harder on our style.
>
>
> In Figure 2: increasing the size of the green star.  We will do this in the camera ready version.
>
>
> Line 365: Citation format needs to be corrected.  This is done

---

### Official Review · Reviewer_qfZf · 2025-10-30

**Soundness:** 3
**Presentation:** 3
**Contribution:** 2
**Rating:** 4
**Confidence:** 4

**Summary:**

This work proposes TALE, a simple pruning method for LLMs that sequentially removes entire transformer layers in a greedy manner based on whether they affect significantly the model validation accuracy on a specific task. This method was developed based on the intuition that sometimes applying the output projection to intermediate layers leads to better performance. Therefore, given the additional residual path of transformer layers, removing entire layers leads to the representations from intermediate layers propagating directly to the output layer. Furthermore, the authors provide a simple information-theoretic analysis on why this phenomenon manifests, visualizing how, an approximate estimate of the, mutual information between representations and the downstream task label changes across layers. Finally, the authors evaluate quite extensively their method on several tasks and architectures.

**Strengths:**

I believe that TALE is an algorithm that can be interesting for practitioners, because of the following reasons

- TALE is very simple to implement; it just requires a task validation set and then multiple inferences to score the contribution of each layer.
- TALE can be applied to arbitrary models, provided that they have a residual path (although the specific residual structure can limit a bit what can be removed)
- Finally, TALE seems to provide performance improvements on a per-task level and also seems to be compatible with finetuning.

**Weaknesses:**

- The task-specific nature of TALE can be limiting; one needs to first have a representative dataset for the downstream task in order to realize the efficiency gains. This is in contrast to more traditional pruning / quantization methods for efficiency that adapt the model on a general dataset and then apply it as is to new tasks. I believe this work could benefit by more comparisons with these set of methods, targeting similar efficiency gains. TALE can also be made "task-agnostic" by optimizing jointly on all tasks as the authors show on Appendix G but the performance is lacking, with the performance being mostly worse than the baseline and the speedup being minor.

- TALE can be computationally expensive due to the greedy search performed, especially when a model is deep. While one could argue this only needs to happen once per task, LLMs now are easily used in multiple tasks via in-context learning / specific prompting, which might lead to excessive computational costs.

- The authors do not  ablate the dependence of TALE on the validation set, which can be a critical hyperparameter. One can imagine that the validation set needs to be sufficiently large in order for the search to succeed and lead to better downstream task accuracy, so its size will be informative for practitioners.

- Finally, the mutual information analysis, while interesting, it is a bit crude as it relies on just linear probes. Furthermore, I am not sure how much additional information it adds, given the results on Appendix C.

**Questions:**

Besides what was mentioned in the weaknesses section, some more questions for the authors are

- Algorithm 1 allows for some performance degradation between steps according to the hyperparameter epsilon, what are typical values for it?
- The baseline performances shown at Table 5 are generally lower than what reported in the respective model cards for Llama 3.1 8B. Any idea on why this is the case? This is important as, for example, the performance on MMLU for Llama 3.1 8B Instruct (the same version that the authors used) is reported as 69.4% here https://huggingface.co/meta-llama/Llama-3.1-8B, which is already better than all of the accuracies reported on Table 5, hence it is unclear what the gains of TALE are in such settings.
- In the first paragraph of the discussion section, the authors mention that their later layer deletion finding challenges prior claims that early layers are redundant and more amenable to removal, without providing any reference for the prior claims. Where was this claim originally mentioned? If anything I would have expected prior claims to be the other way around, as the first layers are the ones that are closest to the raw information of the data.

Overall, at least at the moment, I am on the negative side for this work, primarily due to the missing ablations and comparisons I mentioned in the weaknesses section.

---

> ### Author Response · Authors · 2025-11-17
> **Response to the Reviewer qfZf**
>
> We kindly refer the Reviewer to our general reply above (“Responses to all Reviewers”), where we regrouped responses concerning comparisons with other pruning techniques, computational costs, task specificity of TALE, ablation studies, and Mutual Information.
>
>
> **Re"task specific nature of TALE"** Please see our discussion above (Response to all Reviewers).
>
>
>
> **Re the mutual information analysis** Please refer to our comments above (Response to all Reviewers).
>
>
>
> **Re"Algorithm 1 allows for some performance degradation between steps according to the hyperparameter epsilon, what are typical values for it?"**
>
> In TALE, the user can choose the stopping point depending on the desired threshold accuracy. As illustrated in Figure 2, we observed that in some iterations accuracy could start increasing even after it had started to decrease. For this reason, we allowed the algorithm to continue iterating slightly below the target threshold. Based on our observations, setting $\epsilon=0.08$ was sufficient.
>
>
>
>
> **Re lower baseline performances**
>
> The baseline numbers in Table 5 are lower than those reported on the Hugging Face due to differences in evaluation methodology. The scores in https://huggingface.co/meta-llama/Llama-3.1-8B are computed using an “internal evaluations library”, usually similar to LM-Eval, which selects the highest-probability option among the provided choices rather than evaluating the model’s actual generated outputs. This method tends to inflate scores and compress performance differences; for example, in a two-choice setting, a hallucinated answer still has a 50\% chance of being counted as correct.  We instead report evaluations based on the model’s actual generated outputs, providing a more realistic measure of performance, which also explains why our baseline accuracies are lower than those reported in Hugging Face.
>
>
> **Re:"early layers and later layer importance."**
>
>
>  Thank you for pointing this out. You are right, we expressed this poorly.  What we challenge are the claim that later layers are more important.  We show that we can frequently drop later layers and improve performance.  Claims about the importance of later layers come from here for example:
>
> [1] report that model performance does not drop significantly unless the last two layers are frozen, suggesting the importance of later layers.
>
> [2] find that important feed-forward networks (FFNs) are concentrated in later layers.
>
> [3] show that reasoning accuracy relies heavily on deeper layers.
>
> References
>
>
> [1] https://arxiv.org/pdf/2004.14448
>
> [2]https://aclanthology.org/2023.acl-long.660.pdf
>
> [3]https://arxiv.org/html/2510.02091v1

---

> > ### Comment · Reviewer_qfZf · 2025-11-25
> >
> > I appreciate the effort that the authors put in their rebuttal that addresses some of my concerns. Having said that, I am unfortunately not fully convinced by the method to increase my score.
> >
> > - The comparisons against other baselines are a bit weird; why didn’t the authors equalize the sparsity % between all of the methods? E.g., SparseGPT, Wanda, and SliceGPT operate under 25%-50% sparsity but TALE operates under a 10% sparsity (and therefore can naturally have more flexibility for higher performance)
> > - The highly task specific nature of TALE limits quite a bit the usefulness of the method. Applying it to a mixture of tasks to obtain some generality (as the authors mentioned) is something they sort of did at Appendix J and that did not seem to lead into good results.
> > - The mutual information analysis does not add to the paper (a concern shared by the other reviewers as well)
> > - The concerns about the $\epsilon$ values, the lower baseline performance and importance of early layers are sufficiently addressed.

---

> ### Author Response · Authors · 2025-11-25
> **Response to the Reviewer qfZf**
>
> **Re "Comparisons..."** We thank the reviewer for this observation. We would like to clarify that SLEB is the only directly comparable training-free block-level pruning method to TALE, while SparseGPT/Wanda (weight-level) and SliceGPT (channel-level) represent fundamentally different pruning granularities. We included these baselines as reference points since SLEB compared against them, and they represent the leading training-free methods in the literature.
> Regarding the 10\% sparsity level: this represents SLEB's optimal operating point, where it achieves its best performance before significant degradation at higher sparsity. By comparing at 10\%, we are actually evaluating TALE against SLEB's strongest configuration, not a weakened baseline. The fact that TALE can scale to much higher sparsity levels (as demonstrated in the number of deleted layers in Table 1, where for example on BoolQ using Lucie we deleted 19/32 layers without losing performance) reflects TALE's superior capabilities for task-specific pruning, not an unfair comparison.
>
> **Re "Task specific..."** We respectfully disagree with the premise of this concern. TALE is explicitly designed for task-specific pruning, not general-purpose LLM compression. The task-specific nature is not a limitation but rather the fundamental contribution and intended use case of our method.
> There are many real-world deployment scenarios where task-specific models are preferable or necessary, such as domain-specific applications, edge deployment with constrained resources serving a single application, production systems where models are optimized for specific tasks, multi-agent systems where each agent specializes in a particular task,...
> Moreover, task-specific optimization is a well-established paradigm in LLM deployment: fine-tuning for specific tasks and domain-specific models are widely adopted in practice. Our contribution demonstrates that training-free task-specific pruning is viable and achieves better performance than general-purpose methods when task requirements are known, and synergies even well with fine-tuning.
>
> The comparison in Appendix J was included to demonstrate the inherent trade-off between task-specificity and generality, not as a proposed use case for TALE. Our method intentionally optimizes for the task-specific regime where maximum efficiency gains are possible.
> We position TALE as complementary to general-purpose pruning methods (like SLEB, SparseGPT) rather than a replacement. Users seeking general-purpose compressed models should use those methods; users seeking maximum efficiency for specific tasks should use TALE. Both use cases are valid and important in the landscape of LLM deployment.
>
> **Re "Mutual Information"** We appreciate this feedback. The mutual information analysis was included to provide insight into why TALE achieves performance improvements and why certain layers are more critical than others for task-specific performance. However, we acknowledge that this analysis is tangential to the main thrust of the paper. We are happy to remove or significantly condense this section if multiple reviewers find it does not strengthen the work.
>
> Our main contributions, the TALE algorithm, empirical results, and comparisons to baselines, stand independently of this analysis. The mutual information perspective was exploratory in nature and intended to offer one possible lens for understanding the observed behavior, but we agree it is not essential to the paper's main contributions.
>
> **Re"Other concerns"** We thank the reviewer for acknowledging that these concerns have been satisfactorily addressed.
>
>
>
> We thank the reviewer for their time and thoughtful engagement with our work. We hope our responses have further clarified the reviewer's questions.

---

### Author Response · Authors · 2025-11-17
**Responses to All Reviewers**

We thank all the reviewers for their comments and questions, as well as for their positive findings.  We agree that lightweight implementation, pruning without retraining and compatibility with fine-tuning are important positive features of TALE.

All of you wanted to hear more about comparisons to prior work, including the pros and cons of TALE's task specificity. Most of you also wanted to have more information about computational costs, ablation studies on validation sets used by TALE and the explanation based on Mutual Information.  We address each of these subjects in turn.


**Comparisons**

We agree that a comparison is important. We have now added a comparison in the paper with existing training-free pruning strategies, including [SLEB](https://arxiv.org/pdf/2402.09025), [SliceGPT](https://arxiv.org/pdf/2401.15024), [SparseGPT](https://proceedings.mlr.press/v202/frantar23a/frantar23a.pdf), and [Wanda](https://arxiv.org/pdf/2306.11695). We found that TALE outperforms all of them in accuracy by a significant amount.  The nearest competitors like SparseGPT or SLEB (see below) offer at best very minimal improvements. Below we calculate relative gains or decreases in accuracy with LM eval in the first table and with our evaluation on actual generated output.




Table 1: Accuracies (%) with LM Eval on zero-shot tasks for LLaMA-2-7B and LLaMA-2-13B

| Model       | Method   | Sparsity  | WinoGr   | HellaSwag | ARC-e    | ARC-c    |
| ----------- | -------- | --------- | -------- | --------- | -------- | -------- |
| LLaMA-2-7B | Baseline | 0%        | 69.1     | 76.0      | 74.6     | 46.3     |
|             | SpareGPT | 2:4 (50%) | 65.0     | 58.9      | 60.9     | 34.2     |
|             | Wanda    | 2:4 (50%) | 62.3     | 55.3      | 57.6     | 31.9     |
|             | SliceGPT | 25%       | 63.4     | 54.2      | 58.5     | 34.6     |
|             | SliceGPT | 30%       | 61.3     | 49.6      | 51.8     | 31.2     |
|             | SLEB     | 10%       | 63.1     | 70.2      | 63.7     | 37.7     |
|             | **TALE** | 10%       | **73.1** | **80.0**  | **76.7** | **54.5** |
|  LLaMA-2-13B | Baseline | 0%        | 72.22    | 79.39     | 77.48    | 49.23    |
|             | SpareGPT | 2:4 (50%) | 68.51    | 65.52     | 66.04    | 39.76    |
|             | Wanda    | 2:4 (50%) | 67.01    | 63.09     | 64.31    | 37.80    |
|             | SliceGPT | 25%       | 67.48    | 58.10     | 62.50    | 37.88    |
|             | SliceGPT | 30%       | 65.11    | 52.69     | 56.82    | 35.07    |
|             | SLEB     | 10%       | 66.93    | 74.36     | 71.84    | 41.55    |
|             | **TALE** | 10%       | **76.8** | **83.39** | **80.5** | **53.0** |





Table 2: Accuracies (%) with Our Eval on zero-shot tasks for LLaMA-2-7B and LLaMA-2-13B

| Model       | Method   | Sparsity | ARC-e             | ARC-c             |
| ----------- | -------- | -------- | ----------------- | ----------------- |
| LLaMA-2-7B             | Baseline | 0%       | 51.7              | 40                |
|             | SLEB     | 10%      | 29 (-43.9%)       | 28.8 (-27.5%)     |
|             | **TALE** | 10%      | **62.3** (+25.0%) | **50** (+25.0%)   |
|             | **TALE** | 25%      | **64.8** (+25.3%) | **47.6** (+19.0%) |
| LLaMA-2-13B            | Baseline | 0%       | 73.0              | 54.9              |
|  | SLEB     | 10%      | 43.5 (-40.4%)     | 29.8 (-47.3%)     |
|             | **TALE** | 10%      | **77.3** (+5.9%)  | **64.4** (+17.1%) |
|             | **TALE** | 25%      | **75.3** (+3.2%)  | **64.1** (+16.4%) |



Although general training-free pruning techniques often report acceptable accuracy on standard LM evaluation metrics, their performance remains far below the scores achieved by TALE. Furthermore, the accuracy of their decoded outputs drops sharply when evaluating the model’s actual generations, whereas TALE consistently improves accuracy. This drop occurs because standard LM evaluations typically select the highest-probability option among provided choices, rather than assessing the model’s actual generated outputs, and often exclude hallucinations, which can misrepresent real task performance.

---

> ### Author Response · Authors · 2025-11-17
> **Responses to All Reviewers (2)**
>
> **Computational costs**
>
> Many thanks for your questions about computational costs.
>
>
> We compare the computational cost of the first iteration of TALE and SLEB, two training-free pruning methods evaluated on the ARC-Easy benchmark for LLaMA-2 7B. Like TALE, SLEB uses an iterative greedy algorithm to determine which layers to remove. If TALE runs for $n$ iterations, SLEB also runs for $n$ iterations, although the underlying computation differs substantially. SLEB computes perplexity over the entire WikiText-2 dataset, resulting in approximately $2802.2$ TFLOPs per iteration. In contrast, TALE requires only about $161.3$ TFLOPs per iteration because it operates on a much smaller dataset with shorter sequence lengths. While the computational cost for both methods decreases as more layers are pruned in later iterations, these measurements show that TALE is not at a computational disadvantage relative to the state of the art. As noted above, perplexity computation is itself a nontrivial task requiring significant compute, just as our task-based evaluation does.
>
>
>  While TALE performs a greedy, task-specific search, the actual compute cost is modest. As reported in the revised version, running three full TALE iterations on a task like MMLU (500 validation examples) takes $\approx$ 1 GPU-hour on a single A100, and tasks with shorter sequences require proportionally less time. TALE also removes 3.00\% ± 0.20\% TFLOPs per layer on average, so only a small number of iterations (typically 2–3) are needed to reach common sparsity levels.
> Importantly, this search is performed once per task, while the resulting inference-time savings apply to every future query, making the amortized cost extremely low in practical deployments.
>
>
>
> **Task specificity of TALE**
>
> Reviewers 1 and 3 both asked questions about TALE's task specificity.
> Reviewer 1 had doubts about task specificity.  While TALE is task-specific by design, our experiments show that this is essential for preserving accuracy. We consistently observed that different tasks rely on different subsets of layers: pruning a layer can be catastrophic for one task but beneficial for another.
> This creates a fundamental limitation for general, training-free pruning methods.
> For example, with SLEB-10\% on LLaMA-2-7B, the pruned model  collapses to 29\% accuracy on ARC-easy, 18\% on Winogrande and 28.8\% on ARC-Challenge with our external evaluation on real generated output; a single generally pruned model cannot reliably serve multiple tasks. Other pruning approaches often require retraining to recover performance, but this is costly and still tied to specific training data.
>
>
> Reviewer 3 asks about transferability of TALE accuracies from one task to another. We thank the reviewer for this excellent question.  Indeed when the tasks are close, as in the case of GSM8K and Math500, as we remarked in the paper, deleting a single layer in LLaMA 8B to produce a best Math500 model produces a math model that does better on GSM8K (25\% accuracy as opposed to 15.1\% for the base model). We also looked at transferability of a given task from one language to another.  There the best model for the task in language 1 was also the best model for that task in language 2.  However, when tasks are less similar, the transfer of gains is no longer evident.  The more heterogeneous task B is from task A, the less well the results from a best A model translate to results for B.  We need to investigate this issue further.  One byproduct of that investigation is that TALE will give us a guideline as to how the model sees similarity of tasks.
>
> TALE is general in that it applies to all models and also works in principle with any task and evaluation metric.   Another way to achieve generality will be to apply it to mixtures of tasks and so achieve generality in that way.  We plan to study that in the future.

---

> > ### Author Response · Authors · 2025-11-17
> > **Responses to All Reviewers (3)**
> >
> > **Ablation studies**
> >
> > We thank the reviewers for their insightful questions about validation sets and ablation. TALE requires only a modest validation set to make reliable decisions. We added a new experiment where we vary the size of dataset for each task, ranging from 100 to 1000 examples.
> > As shown in Table below, 100 samples were sufficient to obtain the stable pruning list for LLaMA 3.1 8B on the tested tasks, while for Qwen 2.5 7B the set of layers removed stabilizes with as few as 500 validation examples.
> >
> >
> > Table: Layers removed by TALE for different validation-set sizes across three tasks
> > | Model            | Val Size | Task  | Dropped Layers       |
> > | ---------------- | -------- | ----- | -------------------- |
> > | **Llama 3.1 8B** | 100      | ARC-E | {19, 20, 22, 29, 32} |
> > |                  |          | MMLU  | {21}                 |
> > |                  |          | GSM8K | {3}                  |
> > |                  | 500      | ARC-E | {19, 20, 21, 29, 32} |
> > |                  |          | MMLU  | {21}                 |
> > |                  |          | GSM8K | {3}                  |
> > |                  | 1000     | ARC-E | {19, 20, 21, 29, 32} |
> > |                  |          | MMLU  | {21}                 |
> > |                  |          | GSM8K | {3}                  |
> > | **Qwen 2.5 7B**  | 100      | ARC-E | {22, 27, 28}         |
> > |                  |          | MMLU  | {18, 22, 24, 27, 28} |
> > |                  |          | GSM8K | {19}                 |
> > |                  | 500      | ARC-E | {19, 22, 28}         |
> > |                  |          | MMLU  | {22, 23, 26, 27, 28} |
> > |                  |          | GSM8K | {19}                 |
> > |                  | 1000     | ARC-E | {19, 22, 28}         |
> > |                  |          | MMLU  | {22, 23, 26, 27, 28} |
> > |                  |          | GSM8K | {19}                 |
> >
> >
> >
> > **Mutual Information**
> >
> > We thank the reviewers for their comments and questions about using MI as an explanatory tool.
> >
> > Reviewer 1 worries about the use of probes. They are right. A major challenge of this approach is that it requires to the true probability distributions, which are infeasible to compute. As a result, researchers typically assume a Gaussian distribution [1,2,3] or approximate the probe using a classifier [4,5] or an MLP [6]. In our case, the Gaussian assumption did not fit our datasets. Since we evaluate on QA tasks, we used a trainable classifier to approximate the probes. We redid the section in the revised version of the paper.
> >
> > Reviewer 3 worries about the explanatory nature of MI. There are several options open to analyzing links between layers and performance---mutual information, representational similarity via decomposition or cosine similarity---and all at most indirectly linked to task performance.   Though MI has been used in the literature and seems perhaps the most intuitive of these, we agree that it is not a perfect tool for explaining the behaviors we have noticed.  We are looking at alternative explanatory mechanisms to buttress or complement the MI narrative.
> >
> >
> > [1]https://proceedings.neurips.cc/paper/2018/file/6d0f846348a856321729a2f36734d1a7-Paper.pdf
> >
> >
> > [2] http://proceedings.mlr.press/v38/gao15.pdf
> >
> > [3] https://dl.acm.org/doi/pdf/10.1145/3664647.3680682
> >
> > [4] https://aclanthology.org/2022.cl-1.7.pdf
> >
> > [5] https://arxiv.org/pdf/1610.01644
> >
> > [6] https://proceedings.mlr.press/v80/belghazi18a/belghazi18a.pdf

---

### Note · Authors · 2026-01-05

I have read and agree with the venue's withdrawal policy on behalf of myself and my co-authors.